# Diversity Boosts AI-Generated Text Detection

**Advik Raj Basani** *f20221155@goa.bits-pilani.ac.in*
*Birla Institute of Technology and Science, Goa*

**Pin-Yu Chen** *pin-yu.chen@ibm.com*
*IBM Research*

**Reviewed on OpenReview:** *https://openreview.net/forum?id=WscAU9It1l*

## Abstract

Detecting AI-generated text is an increasing necessity to combat misuse of LLMs in education, business compliance, journalism, and social media, where synthetic fluency can mask misinformation or deception. While prior detectors often rely on token-level likelihoods or opaque black-box classifiers, these approaches struggle against high-quality generations and offer little interpretability. In this work, we propose `DivEye`, a novel detection framework that captures how unpredictability fluctuates across a text using surprisal-based features. Motivated by the observation that human-authored text exhibits richer variability in lexical and structural unpredictability than LLM outputs, `DivEye` captures this signal through a set of interpretable statistical features. Our method outperforms existing zero-shot detectors by up to 33.2% and achieves competitive performance with fine-tuned baselines across multiple benchmarks. `DivEye` is robust to paraphrasing and adversarial attacks, generalizes well across domains and models, and improves the performance of existing detectors by up to 18.7% when used as an auxiliary signal. Beyond detection, `DivEye` provides interpretable insights into why a text is flagged, pointing to rhythmic unpredictability as a powerful and underexplored signal for LLM detection.

## 1 Introduction

Large Language Models (LLMs) have become deeply integrated into daily human workflows, powering applications from personal assistants to academic writing (Alahdab, 2024; Meyer et al., 2023; Lund et al., 2023) and content creation (Hu et al., 2024; Yuan et al., 2022). Their fluency and generalization capabilities make them highly useful, but this same fluency enables a growing number of concerning applications. AI-generated text can now be seamlessly inserted into essays, news articles, legal briefs, scientific abstracts, and social media posts, often without detection (De Giorgio et al., 2025; Papageorgiou et al., 2024; Telenti et al., 2024; Törnberg et al., 2023).

As LLM-generated outputs grow more sophisticated and human-like, detecting them has become an increasingly difficult challenge (Abdali et al., 2024; Gameiro et al., 2024; Wu et al., 2025; Zhang et al., 2024). Reliable AI-text detection is crucial for mitigating risks such as misinformation, AI-assisted academic dishonesty, professional misconduct, and the inadvertent suppression of authentic human writing. Traditional approaches to this problem rely on supervised detectors (Shukla et al., 2024; Tolstykh et al., 2024; Wang et al., 2024b) trained on annotated datasets of AI and human-authored text. These models often incorporate rich features, ranging from stylometry and structure to information-theoretic metrics, and achieve high performance within the domain they were trained on. However, such methods struggle to generalize to unseen models or domains (Doughman et al., 2024; Gameiro et al., 2024), especially as new LLMs are frequently released. In contrast, zero-shot detectors (Bao et al., 2024; Gehrmann et al., 2019; Mitchell et al., 2023; Wang et al., 2024a) offer a promising alternative by avoiding model-specific training. These approaches either extract statistical cues from language model probability distributions or use LLMs themselves as

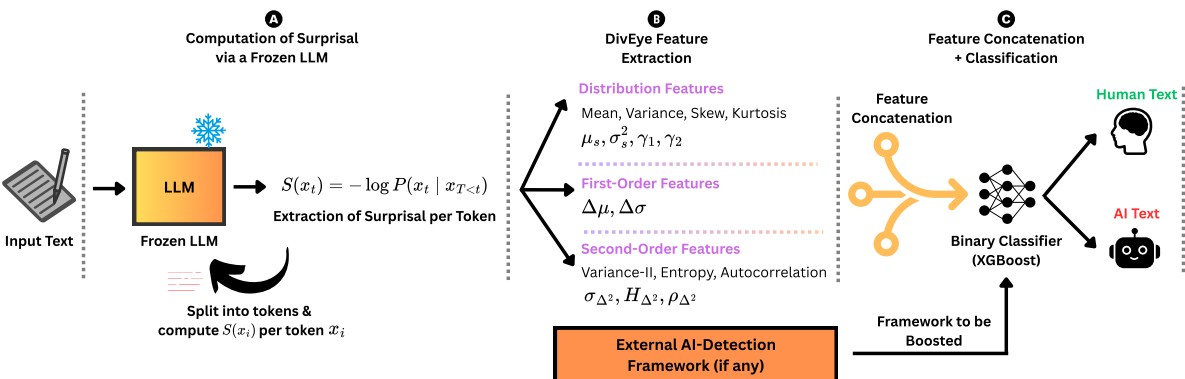

Figure 1: Overview of `DivEye`. `DivEye` extracts diversity-based features (see Section 3, Equation 6) from token-level surprisal patterns. These features can be used in two ways: (1) as a standalone detector, or (2) as an enhancement to existing detectors, improving their performance.

inference-time detectors, enabling model-agnostic detection at scale. Given the increasing deployment of unknown or fine-tuned LLMs in the wild, zero-shot detection has become an essential tool for maintaining platform integrity and addressing the forensic needs of AI-era communication.

**Contributions.** We introduce `DivEye`[1], a lightweight classifier trained on features extracted from off-the-shelf LLMs in a zero-shot manner. These features capture diversity-based statistics of token-level surprisal, which we leverage to improve AI-text detection. Our approach focuses on capturing the distributional irregularities in AI-generated text that arise from differences in the generative process compared to human writing.

- *Zero-shot diversity detection:* We propose `DivEye`, a lightweight classifier trained on zero-shot features derived from token-level surprisal diversity metrics. These metrics capture fluctuations and patterns that reflect the constrained and often repetitive generation process of LLMs. We provide a principled motivation for each feature, connecting them to known properties of human vs. machine text generation, and demonstrate how `DivEye` can improve AI-text detection using these features.

- *Language & Model-agnostic detection:* `DivEye` leverages zero-shot features, requiring no access to the generator model's internals or any fine-tuning. It operates purely on token probability sequences from an off-the-shelf language model and generalizes across different languages and model families.

- *Complementary to existing detectors:* We show that `DivEye` captures statistical patterns that are distinct from those used by traditional detectors, which often rely on fine-tuned language representations or classifier-based signals. When combined with these approaches, `DivEye` significantly boosts overall robustness, particularly against challenging high-quality generations and paraphrased adversarial examples.

- *Strong generalization across domains and attacks:* Extensive evaluations across three benchmarks and varied testbeds reveal that `DivEye` not only achieves state-of-the-art accuracy in standard settings but also remains robust when tested on unseen domains and language models.

## 2 Background and Problem Formulation

The emergence of LLM has led to a new era of machine-generated text that can closely mimic human writing across a range of tasks. These models are trained to approximate the true conditional distribution of natural language, denoted as $P_{\text{human}}(x_t \mid x_{<t})$, by learning from massive corpora of human-written text (Chen et al., 2024; Lu et al., 2025). The LLM's learned distribution is represented as $P_{\text{LLM}}(x_t \mid x_{<t})$, and during inference, the model generates text by sampling tokens sequentially from this distribution. While modern LLMs achieve remarkable fluency, they still constitute an imperfect approximation: $P_{\text{LLM}} \neq P_{\text{human}}$

---

[1]The code of our method and experiments is available at `https://github.com/IBM/diveye`.

in general. At inference time, an LLM selects tokens by sampling from this learned distribution (Zhou et al., 2024), which remains an approximation of the true distribution that governs human text generation (Ippolito et al., 2020; Jones et al., 2024). This approximation gap, subtle as it may be, is the crux of AI text detection.

From a theoretical standpoint, prior works (Ghosal et al., 2023; Sadasivan et al., 2025) highlight the fundamental limitations of AI text detection: as generative models approach the ideal of human-like language modeling, distinguishing their outputs from real text becomes increasingly difficult, if not impossible. Yet, as Chakraborty et al. (2023) point out, even models arbitrarily close to optimal remain statistically detectable under certain conditions, particularly when multiple samples or robust features are available. This theoretical detectability provides a foundation for practical detection methods that capitalize on the subtle imperfections in current LLM outputs.

In practice, existing detection approaches fall into two broad categories: watermarking and zero-resource detection. Watermarking techniques (Block et al., 2025; Gloaguen et al., 2025; Kirchenbauer et al., 2024; Liang et al., 2024; Liu et al., 2024a) embed distinct patterns in generated text but necessitate access to model internals or fine-tuning capabilities, rendering them unsuitable for black-box or adversarial settings, as well as for practical cases of watermark-free AI text detection. In contrast, zero-resource detection methods require no prior knowledge of the target model, instead relying on statistical or learned discrepancies between human and AI text. These methods can be further categorized as statistical and training-based approaches.

**Statistical / Zero-shot detection methods** refers to identifying AI-generated text without task-specific training, either by leveraging LLM probability cues or prompting LLMs directly as detectors. For example, methods like Entropy (Lavergne et al., 2008), LogRank (Ghosal et al., 2023), DetectGPT (Bao et al., 2024; Mitchell et al., 2023), and Binoculars (Hans et al., 2024) use off-the-shelf LLMs to evaluate the consistency of token predictions under masked or perturbed inputs. These methods assume that AI-generated texts are sampled from a narrower, more concentrated conditional probability distribution than human writing, resulting in greater token-level confidence and reduced lexical diversity.

**Training-based / Fine-tuned detection methods** (Chen et al., 2023; Mao et al., 2024; Hu et al., 2023) train classifiers, such as fine-tuned transformers on a labeled corpora of human and AI text. While these models can be accurate, they often fail to generalize across domains or against adversarial paraphrasing, especially when trained on specific generators or prompts. We discuss all related works in more detail in Appendix A.

Despite significant progress, no existing method fully resolves the problem of detecting AI-generated text in the wild. Our work addresses this gap by approaching the problem from a new angle: instead of analyzing individual token probabilities (Solaiman et al., 2019) in isolation, we propose to measure statistical diversity over token sequences, quantifying how text fluctuates in its use of surprising or predictable tokens. This provides a more global signature of the generative process that is robust to paraphrasing, domain shifts, and even partial text corruption.

## 3 `DivEye`: Methodologies

`DivEye` is built on the central observation that fluctuations in token-level surprisal provide a strong signal for distinguishing machine- and human-generated text. By systematically analyzing the statistical variation of surprisal across a sequence, `DivEye` captures distributional and temporal patterns that go beyond traditional likelihood-based metrics. The name `DivEye` thus reflects our method's focus on diversity-aware analysis of language generation behavior.

### 3.1 Design Hypothesis

A central challenge in detecting AI-generated text (Ghosal et al., 2023; Sadasivan et al., 2025) lies in the fact that current models, though fluent, often prioritize coherence and consistency at the cost of variability and unpredictability. By contrast, human writers naturally introduce irregularities, such as unexpected lexical choices or structural shifts, that make their text inherently more diverse.

**Our hypothesis is that human-written text inherently exhibits greater stylistic diversity and unpredictability than AI-generated text.** In everyday writing, humans make creative, spontaneous choices, sometimes using unexpected words or phrases, that introduce bursts of surprise amid more routine language. Our approach centers on the premise that AI-generated text, despite its fluency, often lacks the inherent diversity observed in human-written language. This divergence stems from the fundamental objective of LLMs: to maximize the likelihood of generated sequences within their learned probability distributions (Park & Choi, 2024). Consequently, AI-generated text tends to exhibit a higher degree of predictability, resulting in lower variability and surprisal compared to human-authored content. We support this hypothesis through both intuitive reasoning and empirical evidence, as detailed in Remark 1.

---

**Remark 1** Consider a text sequence $X = (x_1, x_2, \ldots, x_n)$ generated either by a human or by a language model $M$. The language model defines a probability distribution $P_M(X) = \prod_{t=1}^{n} P_M(x_t \mid x_{<t})$ where each token is chosen to maximize overall likelihood.

Humans, however, produce language through a complex, multi-layered cognitive process that balances informativeness, creativity, and contextual appropriateness, rather than strictly maximizing statistical likelihood.

Formally, the surprisal of token $x_t$ under model $M$ is defined as:

$$S_M(x_t) = -\log P_M(x_t \mid x_{<t})$$

Since $M$ is trained to assign high probability to plausible continuations, its outputs tend to minimize surprisal on average, implying that maximum likelihood generation compresses diversity:

$$\mathbb{E}_{X \sim P_M}[S_M(x_t)] \leq \mathbb{E}_{X \sim P_H}[S_M(x_t)]$$

where $P_H$ denotes the distribution of human-generated text.

Similarly, human language exhibits higher variance in surprisal due to spontaneous creative choices, idiomatic expressions, and stylistic variation, causing:

$$\text{Var}_{X \sim P_M}[S_M(x_t)] < \text{Var}_{X \sim P_H}[S_M(x_t)]$$

We validate this theoretical intuition through empirical experiments detailed below, which confirm statistically significant differences in surprisal and diversity metrics between human-written and AI-generated texts.

We collect 200 human-written essays and 200 GPT-4-Turbo-generated essays on comparable topics, provided by BiScope (Guo et al., 2024). For each essay, we computed the token-level surprisal scores using a fixed language model evaluator (GPT-2) and then calculated the mean and variance of these surprisal values per essay. Figure 2a shows the histogram of mean surprisal scores across the two sets, while Figure 2b displays the histogram of surprisal variances. The human-written texts exhibit a noticeably wider spread and heavier tails in both metrics, indicating greater unpredictability and stylistic variability. In contrast, the AI-generated essays cluster around lower mean surprisal and exhibit significantly lower variance. These results empirically confirm our theoretical claim: **human language inherently reflects higher diversity and surprise, whereas AI-generated language, optimized for likelihood, tends toward more predictable and homogeneous patterns.**

---

Rather than treating token-level surprisal in isolation, `DivEye` analyzes how it varies across an entire text to capture higher-level stylistic patterns. By extracting global statistical features from surprisal sequences, our method reveals differences in the rhythm and variability of unpredictability, traits that distinguish human writing from the more uniform outputs of LLMs, as illustrated by the clear class separation in predicted probabilities shown in Figure 3.

These features are theoretically grounded in the Uniform Information Density (UID) hypothesis, which posits that optimal language speakers strives to distribute information uniformly Jaeger & Levy (2006). However,

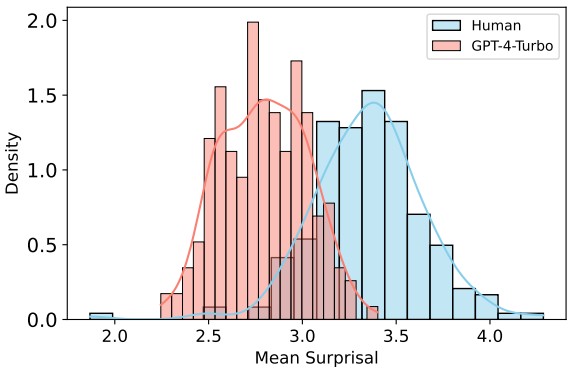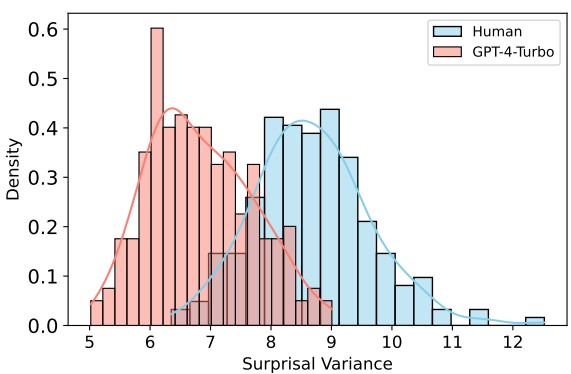

(a) Mean Surprisal Distribution

(b) Surprisal Variance Distribution

Figure 2: Distribution of token-level surprisal metrics for human-written vs. GPT-4-Turbo-generated essays. The left plot shows the histogram of mean surprisal per essay, while the right plot shows the histogram of surprisal variance. Human-written texts exhibit higher dispersion and heavier tails in both distributions, suggesting greater linguistic unpredictability and stylistic diversity. In contrast, GPT-4-Turbo outputs are more concentrated and predictable, aligning with the likelihood-maximization objective of language models.

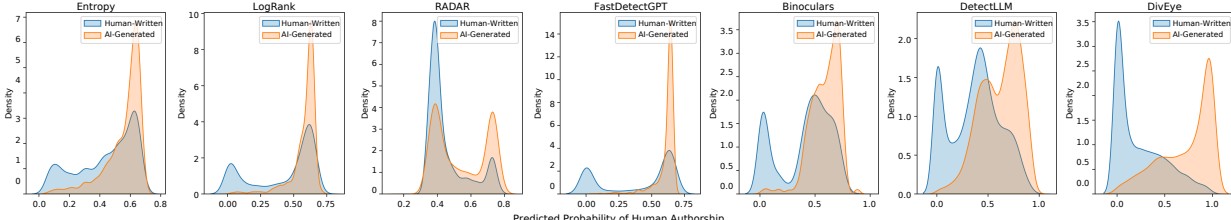

Figure 3: Distributions of predicted class probabilities for diverse AI-text detectors. Trained and evaluated on Testbed 4 of the MAGE benchmark, `DivEye` shows stronger separation between human-written and Label 1 AI-generated texts, indicating greater confidence and discriminative power.

while human speakers aim for this efficiency, empirical evidence shows that natural language remains highly nonstationary (Piantadosi et al., 2011) and exhibits characteristic "burstiness" due to semantic shifts (Altmann et al., 2009). In contrast, LLMs, driven by likelihood-maximization objectives, often approximate the theoretical UID optimum too strictly, resulting in unnaturally flat probability curves devoid of these organic fluctuations (Holtzman et al., 2020). Consequently, we introduce first-order and second-order derivatives of surprisal, as essential metrics to quantify this 'rhythmic unpredictability', capturing the distinct structural variance that separates human writings from LLMs.

## 3.2 Mathematical underpinning of `DivEye`

To robustly distinguish AI-generated text from human-written text, it is insufficient to rely solely on a single measure such as perplexity (Xu et al., 2024). Perplexity summarizes average token likelihood, but overlooks how unpredictability fluctuates within a text. To better capture these patterns, `DivEye` computes higher-order statistical features over surprisal sequences, revealing structural signals beyond aggregate likelihood.

**Surprisal.** Human language is inherently diverse and unpredictable, balancing consistent patterns with bursts of creativity, often introducing novel expressions, grammatical deviations, and stylistic variation. These deviations result in varying levels of token predictability, which can be quantified using surprisal (Kuribayashi et al., 2025) - a well-known information-theoretic measure defined as the negative log-probability of a token under a language model:

$$S(x_t) = -\log P(x_t \mid x_1, x_2, \ldots, x_{t-1})$$

Given a text sequence $X = \{x_1, x_2, \ldots, x_n\}$, surprisal measures how "unexpected" each token is in context. It can be computed directly from a model's log-probabilities, providing a principled way to quantify the local unpredictability of text.

Rather than examining individual token surprisals in isolation, we summarize their behavior through aggregate metrics. The mean surprisal serves as a coarse indicator of how "expected" a text is on average: Lower values suggest higher conformity to the model's learned distribution, whereas higher values point to more frequent unpredictability. However, as stated before, human writing is not merely unpredictable in aggregate; it also exhibits fluctuations in predictability that correspond to stylistic variation, topic shifts, or bursts of creativity. This motivates analyzing not just the mean but also the variance of surprisal, which captures the extent of variation in token-level surprise throughout the text. Formally, this can be represented as:

$$\text{Mean: } \mu_S = \frac{1}{n} \sum_{t=1}^{n} S(x_t); \quad \text{Variance: } \sigma_S^2 = \frac{1}{n-1} \sum_{t=1}^{n} (S(x_t) - \mu_S)^2 \tag{1}$$

**Mean and Variance are not sufficient.** While mean and variance capture the central tendency and spread of surprisal values, they overlook deeper structural signals that differentiate human and AI text. AI-generated text is optimized for consistency, producing more symmetrical distributions centered around high-probability tokens (Ippolito et al., 2020). Skewness ($\gamma_1$) quantifies this asymmetry: a positive skew suggests the presence of rare, surprising tokens typically found in human writing. Similarly, kurtosis ($\gamma_2$) captures the frequency of extreme deviations from the norm. A high kurtosis indicates heavy-tailed behavior, another hallmark of authentic, stylistically diverse writing. These higher-order moments allow `DivEye` to detect subtle irregularities and stylistic outliers that can be missed by detectors focusing only on average behavior.

$$\text{Skewness: } \gamma_1 = \frac{1}{n} \sum_{t=1}^{n} \left( \frac{S(x_t) - \mu_S}{\sigma_S} \right)^3; \quad \text{Kurtosis: } \gamma_2 = \frac{1}{n} \sum_{t=1}^{n} \left( \frac{S(x_t) - \mu_S}{\sigma_S} \right)^4 - 3. \tag{2}$$

**Static metrics still miss temporal structure.** While static surprisal statistics (mean, variance, skewness, kurtosis) describe the overall distribution of token-level unpredictability, they fail to capture how this unpredictability evolves throughout a sequence, a key trait distinguishing human and AI-generated text. To model these temporal dynamics, we compute the first-order difference $\Delta S_t = S(x_t) - S(x_{t-1})$, which reflects immediate changes in surprisal. The mean ($\Delta\mu$) and variance ($\Delta\sigma^2$) of $\Delta S_t$ quantify the typical magnitude and variability of these shifts, capturing stylistic volatility such as abrupt topic or tone changes commonly found in human writing.

We further analyze the second-order difference $\Delta^2 S_t = \Delta S_t - \Delta S_{t-1}$, which tracks fluctuations in the rate of change of surprisal. From this sequence, we extract three metrics: (1) variance ($\sigma_{\Delta^2}^2$), to capture the extent of rapid or erratic stylistic transitions; (2) entropy ($\mathcal{H}_{\Delta^2}$), which reflects the irregularity of these transitions; and (3) autocorrelation ($\rho(\Delta^2 S_t)$), which measures whether bursts of unpredictability cluster together, often indicative of structured human creativity. These second-order metrics reveal rhythmic and non-stationary patterns in human text that are typically absent in the more homogeneous output of LLMs, providing a richer signal for robust AI-text detection. Mathematically, these can be defined as:

$$\Delta S_t = S(x_t) - S(x_{t-1}), \qquad \Delta\mu = \frac{1}{n-1} \sum_{t=2}^{n} \Delta S_t, \quad \Delta\sigma^2 = \frac{1}{n-2} \sum_{t=2}^{n} (\Delta S_t - \mu_\Delta)^2 \tag{3}$$

$$\Delta^2 S_t = \Delta S_t - \Delta S_{t-1}, \qquad \sigma_{\Delta^2}^2 = \frac{1}{n-3} \sum_{t=3}^{n} (\Delta^2 S_t - \mu_{\Delta^2})^2, \quad \mathcal{H}_{\Delta^2} = -\sum_b p_b \log p_b, \tag{4}$$

$$\rho(\Delta^2 S_t) = \frac{\mathbb{E}\left[(\Delta^2 S_t - \mu_{\Delta^2})(\Delta^2 S_{t+1} - \mu_{\Delta^2})\right]}{\sigma_{\Delta^2}^2} \tag{5}$$

where $\mu_{\Delta^2}$ is mean of second-order differences, and $p_b$ is the empirical probability of a value falling into bin $b$ after discretizing $\Delta^2 S_t$ for entropy computation. We provide empirical validation of these temporal features and their individual contributions to detection performance in Appendix B.

**Combinations.** Collectively, `DivEye`, formalized as ($\mathcal{D}$) in Equation 6, encapsulates critical aspects of text generation that distinguish human creativity from algorithmically generated predictability, thereby serving as a robust basis for our detection framework.

$$\mathcal{D} = \{\underbrace{\mu_s, \sigma_s^2, \gamma_1, \gamma_2}_{\text{Distribution}} \oplus \underbrace{\Delta\mu, \Delta\sigma^2}_{\text{1st-Order}} \oplus \underbrace{\sigma_{\Delta^2}^2, H_{\Delta^2}, \rho_{\Delta^2}}_{\text{2nd-Order}} \tag{6}$$

Here, $\mathcal{D}$ is a vector of statistical features with a dimension of 9, including distributional properties, first-order differences, and second-order differences of the text. We can apply any autoregressive LLM to generate these feature vectors by passing the text tokens through the model to compute the features listed above, which are then concatenated into the final vector $\mathcal{D}$. We train a binary classifier using `DivEye` features, optionally combined with predictions from an AI-text detector. Implementation details are further explained in Algorithm 1 and Appendix C .

**`DivEye` as a booster.** Existing detectors, whether fine-tuned classifiers or zero-shot LLM-based methods, primarily rely on semantic or surface-level cues, and often falter against high-quality adversarial examples that closely mimic human writing. `DivEye` offers a complementary signal by capturing statistical and temporal patterns of token-level unpredictability that are orthogonal to traditional features. We integrate `DivEye` into both settings by augmenting detector outputs with its feature vector and training a lightweight meta-classifier (e.g., XGBoost (Chen & Guestrin, 2016) or Random Forest (Breiman, 2001)) over the combined representation. Empirically, we find that this fusion significantly boosts performance, particularly on adversarial and out-of-distribution examples, without requiring retraining or modification of the original model.

## 4 Experiments

### 4.1 Experimental Settings

**Datasets.** We evaluate our `DivEye` framework across a comprehensive suite of datasets that encompass a wide range of generative models, domains, and adversarial strategies. Our primary benchmark is the RAID dataset (Dugan et al., 2024), which consists of carefully crafted adversarial examples designed to evade standard detectors. To assess robustness under diverse generation conditions, we also evaluate on the MAGE benchmark (Li et al., 2024), which spans eight distinct testbeds targeting various domains (e.g., Yelp (Zhang et al., 2015), XSum (Narayan et al., 2018), SciXGen (Chen et al., 2021a), CMV (Tan et al., 2016)) and generator families (e.g., GPT (Radford et al., 2019), OPT (Zhang et al., 2022), Bloom (et al., 2023)). This granular evaluation allows us to isolate and quantify the contribution of diversity metrics across specific domains and model types. Details about each testbed in RAID & MAGE are discussed in Appendix H.

Additionally, we incorporate HC3 (Guo et al., 2023), a large-scale, heterogeneous corpus of human and machine text, which includes both English and Chinese instances of human and AI-generated Q&A data. The inclusion of HC3 enables us to probe cross-linguistic generalization of our method.

**Baselines.** We compare `DivEye` with various baselines, including both traditional statistical detectors and recent fine-tuned models. These include RADAR (Hu et al., 2023), LogRank (Ghosal et al., 2023), Entropy (Lavergne et al., 2008), FastDetectGPT (Bao et al., 2024), DetectLLM (Su et al., 2023), Binoculars (Hans et al., 2024), RAiDAR (Mao et al., 2024), OpenAI Detector (Solaiman et al., 2019), Longformer (Beltagy et al., 2020), and BiScope (Guo et al., 2024). These baselines cover a range of techniques, from token-level likelihood-based ranking to transformer-based classification. Additionally, we evaluate our framework against several other open-source detectors listed on the RAID leaderboard, ensuring a fair and broad comparison with state-of-the-art public tools across multiple detection paradigms.

**Implementation Details & Metrics.** Unless stated otherwise, we use GPT-2 to compute all `DivEye` feature vectors. Regardless, we studied the effect of `DivEye` with different LLMs as base models and summarized the results in Section 4.6. In score-only detection scenarios, predictions are based solely over concatenated `DivEye` features. For both standalone and boosted setups, we train a lightweight XGBoost (Chen & Guestrin, 2016) classifier as a meta-model, using only `DivEye` features in the former, and concatenating them with

Table 1: Performance of zero-shot methods on 6 diverse testbeds from MAGE. The OOD settings examine the detection capability on texts from unseen domains or texts generated by new LLMs.

| Settings | Methods | HumanAcc | MachineAcc | AvgAcc | AUROC |
|---|---|---|---|---|---|
| **Testbed 2,3,4: In-distribution detection** | | | | | |
| Arbitrary-domains & Model-specific (GPT-J [105]) | LogRank | 58.81% | 63.94% | 61.38% | 0.67 |
| | Entropy | 76.43% | 76.84% | 76.64% | 0.83 |
| | DetectLLM | 66.36% | 62.07% | 64.21% | 0.72 |
| | FastDetectGPT | 62.31% | 50.49% | 56.4% | 0.59 |
| | Binoculars | 60.11% | 65.22% | 62.67% | 0.69 |
| | BiScope | 89.62% | 84.86% | 87.24% | 0.93 |
| | DivEye | 90.63% | 88.56% | **89.60%** | **0.97** |
| Fixed-domain (WP [35]) & Arbitrary-models | LogRank | 89.61% | 56.15% | 72.88% | 0.76 |
| | Entropy | 85.96% | 60.4% | 73.18% | 0.78 |
| | DetectLLM | 88.54% | 80.77% | 84.66% | 0.91 |
| | FastDetectGPT | 87.25% | 54.08% | 70.67% | 0.76 |
| | Binoculars | 80.80% | 62.07% | 71.44% | 0.77 |
| | BiScope | 91.78% | 95.27% | 93.53% | 0.94 |
| | DivEye | 92.22% | 96.88% | **94.55%** | **0.99** |
| Arbitrary-domains & Arbitrary-models | LogRank | 84.91% | 44.47% | 64.69% | 0.68 |
| | Entropy | 75.68% | 50.04% | 62.86% | 0.67 |
| | DetectLLM | 64.74% | 69.02% | 66.88% | 0.75 |
| | FastDetectGPT | 93.65% | 41.73% | 67.69% | 0.7 |
| | Binoculars | 76.1% | 54.89% | 65.49% | 0.71 |
| | BiScope | 91.54% | 58.70% | 75.12% | 0.86 |
| | DivEye | 73.72% | 82.57% | **78.15%** | **0.88** |
| **Testbed 5,6,8: Out-of-distribution detection** | | | | | |
| Unseen Models (BLOOM-7B [31]) | LogRank | 85.84% | 19.82% | 52.89% | 0.52 |
| | Entropy | 77.56% | 34.74% | 56.15% | 0.59 |
| | DetectLLM | 67.85% | 58.5% | 63.18% | 0.68 |
| | FastDetectGPT | 94.57% | 13.81% | 54.19% | 0.54 |
| | Binoculars | 76.10% | 54.89% | 65.50% | 0.71 |
| | BiScope | 76.72% | 50.47% | 63.60% | 0.72 |
| | DivEye | 74.75% | 77.06% | **75.91%** | **0.86** |
| Unseen Domains (WP [35]) | LogRank | 88.57% | 49.8% | 69.19% | 0.74 |
| | Entropy | 78.5% | 58.16% | 68.33% | 0.74 |
| | DetectLLM | 74.15% | 71.52% | 72.34% | 0.79 |
| | FastDetectGPT | 95.99% | 47.17% | 71.58% | 0.74 |
| | Binoculars | 78.93% | 67.8% | 73.37% | 0.8 |
| | BiScope | 80.1% | 78.3% | 79.2% | 0.86 |
| | DivEye | 94.64% | 84.53% | **89.59%** | **0.97** |
| Unseen Domains & Unseen Models | LogRank | 83.87% | 43.95% | 63.91% | 0.68 |
| | Entropy | 74.93% | 50.18% | 62.55% | 0.66 |
| | DetectLLM | 63.66% | 67.40% | 65.53% | 0.73 |
| | FastDetectGPT | 93.38% | 41.50% | 67.44% | 0.70 |
| | Binoculars | 77.85% | 69.39% | 73.62% | 0.81 |
| | BiScope | 86% | 82.58% | **84.24%** | **0.92** |
| | DivEye | 69.75% | 83.22% | 76.49% | 0.87 |

the original detector's prediction scores in the latter. Each testbed in MAGE & RAID provides predefined training and test sets, which we use for model training and evaluation. We evaluate all models using Average Accuracy (AvgAcc), AUROC, TPR@FPR=5% and F1 score to capture overall, threshold-independent, and balanced performance, respectively. For all methods, we independently optimize any required decision thresholds for AUROC and for TPR@FPR=5%, ensuring that each model is evaluated under its most favorable threshold for each metric.

## 4.2 Performance of DivEye

We evaluate DivEye across a wide range of challenging testbeds to assess its robustness and adaptability to both domain and model shifts. Table 1 presents the performance of DivEye on six distinct testbeds from the MAGE benchmark (Li et al., 2024): three in-distribution and three out-of-distribution. Across all testbeds,

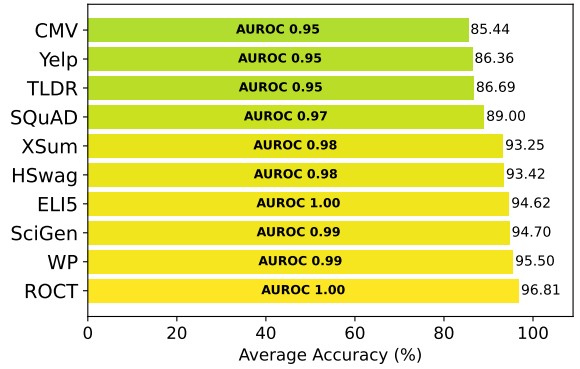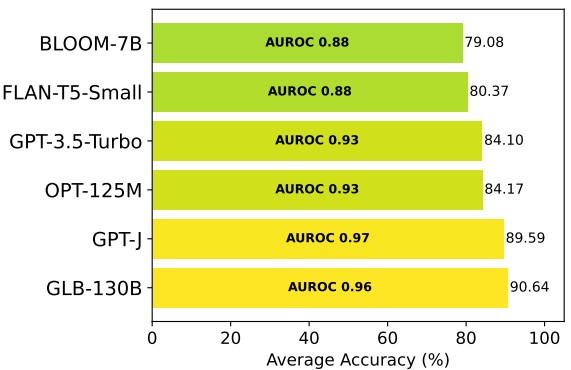

Figure 4: (a) Performance of `DivEye` across different domains, generated by `GPT-J-6B`. (b) Performance of `DivEye` across various generator models. Results are based on the MAGE benchmark.

Table 2: Performance of zero-shot and open-source finetuned methods on RAID. Results are aggregated over 8 domains, 12 models, and 4 decoding strategies. $\delta$ denotes difference in AUROC from benchmark leader.

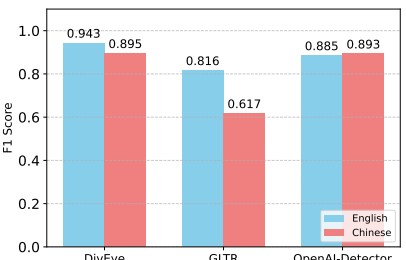

| Frameworks | Type | TPR@ FPR=5% | AUROC | $\delta$ |
|---|---|---|---|---|
| e5-small-lora [30] | Fine-tuned | 93.9% | 0.986 | -0% |
| **DivEye (Ours)** | **Classifier (zero-shot)** | **93.63%** | **0.984** | **-0.20%** |
| Desklib AI [28] | Fine-tuned | 94.9% | 0.973 | -1.32% |
| SuperAnnotate [95] | Fine-tuned | 70.3% | 0.910 | -7.71% |
| Binoculars [46] | Zero-shot | 79.0% | 0.844 | -14.40% |
| RADAR [49] | Fine-tuned | 65.6% | 0.819 | -16.94% |
| GLTR [39] | Zero-shot | 59.7% | 0.727 | -26.27% |

Figure 5: F1 scores on HC3 show that `DivEye` outperforms GLTR [39] and OpenAI-Detector [93], with strong results across English and Chinese.

`DivEye` consistently achieves superior AUROC of 0.92 on average and AvgAcc compared to existing zero-shot baselines, showcasing its ability to generalize effectively to both seen and unseen generation settings. We demonstrate that human-written and machine-generated text can be distinguished based on the hypothesis outlined in Section 3. We further include the corresponding TPR@FPR=5% performance in Table 5.

Table 2 benchmarks `DivEye` on the RAID dataset (Dugan et al., 2024), which includes a suite of diverse models, domains, attacks, and decoding strategies. `DivEye` outperforms a diverse set of zero-shot methods by 26.11% and matches the performance of generative detection baselines, reaffirming its robustness to evasive generation strategies. Figures 4, 7 & 8 demonstrate the performance of `DivEye` across different domains and generator models, achieving competitive AUROC of 0.98 and 0.93, respectively. These results highlight `DivEye`'s stability and high performance across heterogeneous scenarios, underscoring its domain and model-agnostic nature.

Moreover, Appendix D.5 reports `DivEye`'s detection rates on all major models, including GPT-3.5-Turbo (Brown et al., 2020) and GPT-4o (et al., 2024b), Claude-3-Opus and Sonnet (Anthropic), as well as Gemini-1.0-Pro (et al., 2025), demonstrating highly competitive accuracies across the board. Collectively, these results confirm that `DivEye` provides a robust and adaptable foundation for detecting AI-generated text.

### 4.3 Robustness to Adversarial Attacks and Multilingual Text

To evaluate robustness, we assess `DivEye` on a diverse set of adversarial attacks, including paraphrasing attacks from the MAGE dataset and transformation-based jailbreak attacks from the RAID benchmark. Table 3 shows that `DivEye` consistently achieves strong detection performance under these challenging settings. On the MAGE benchmark, `DivEye` outperforms the fine-tuned Longformer baseline in both average

Table 4: Integration with `DivEye` consistently boosts performance across detectors, particularly on diverse domains (Testbed 4) and paraphrasing attacks (Testbed 7).

| Methods | HumanAcc | MachineAcc | AvgAcc | AUROC | $\delta$: Boost |
|---|---|---|---|---|---|
| **Testbed 4: Arbitrary Domains & Arbitrary Models** | | | | | |
| RADAR | 47.74% | 74.86% | 61.30% | 0.62 | - |
| DetectLLM | 64.74% | 69.02% | 66.88% | 0.75 | - |
| FastDetectGPT | 93.65% | 41.73% | 67.69% | 0.7 | - |
| Binoculars | 76.1% | 54.89% | 65.49% | 0.71 | - |
| BiScope | 91.54% | 58.70% | 75.12% | 0.86 | |
| DivEye | 73.72% | 82.57% | 78.15% | 0.88 | - |
| DivEye + RADAR | 74.69% | 85.31% | 80% | 0.90 | **18.7%** |
| DivEye + DetectLLM | 75.44% | 84.23% | 79.34% | 0.9 | 12.96% |
| DivEye + FastDetectGPT | 79.42% | 83.90% | 81.66% | 0.91 | 13.97% |
| DivEye + Binoculars | 69.81% | 83.47% | 76.64% | 0.87 | 11.15% |
| DivEye + BiScope | 80.69% | 88.31% | **84.5%** | **0.93** | 9.38% |
| **Testbed 7: Paraphrasing Attacks** | | | | | |
| BiScope | 48.80% | 89.79% | 69.30% | 0.81 | - |
| DivEye | 69.75% | 83.22% | 76.49% | 0.87 | - |
| DivEye + BiScope | 65.38% | 90.84% | **78.11%** | **0.89** | **8.81%** |

accuracy and AUROC by 10.15% and 0.11 respectively. On the RAID benchmark, which reports only accuracy, `DivEye` achieves competitive results across a range of adversarial perturbations, outperforming several zero-shot detectors, most notably surpassing Binoculars by 11.2%. A more detailed breakdown of performance by individual attack type is provided in Appendix E.1. We also test `DivEye`'s robustness to diverse adversarial scenarios, including character- and word-level perturbations, paraphrasing via commercial tools, prompt obfuscations, and distributional shifts, and find that it consistently achieves exceptional detection performance; a consolidated overview is provided in Appendix E.

We further evaluate `DivEye`'s multilingual generalizability using both English and Chinese splits of the HC3 dataset. Figure 5 illustrates that `DivEye` performs consistently well and has higher F1 scores across both languages using `GPT-2` (Radford et al., 2019) & `GPT-2-Chinese` (CKIPLAB, 2024) for English and Chinese respectively. This suggests that surprisal-based statistical features are not heavily language-specific and can generalize across languages.

### 4.4 Efficiency Analysis

In addition to accuracy, we analyze the computational efficiency of `DivEye`. Figure 6b illustrates the latency of our method, showing that `DivEye` requires as little as 0.01 seconds per sample while outperforming several fine-tuned and zero-shot detectors, achieving up to a 2971× speedup compared to RAiDAR. Be-

Table 3: Performance of `DivEye` and baselines on adversarial benchmarks, MAGE & RAID. * - RAID reports TPR@FPR=5% instead of AvgAcc.

| Settings | Methods | AvgAcc | AUROC |
|---|---|---|---|
| **[MAGE] Testbed 8: Paraphrasing Attack** | | | |
| Paraphrased via GPT-3.5-Turbo | Longformer [11] | 69.34% | 0.76 |
| | BiScope [45] | 69.30% | 0.81 |
| | DivEye (Ours) | **76.49%** | **0.87** |
| **[RAID] Adversarial Attacks*** | | | |
| Paraphrase, Whitespace, Misspelling, Homoglyph, Article Deletion & more | Desklib AI [28] | 91.2% | 0.948 |
| | e5-small-lora [30] | 85.7% | 0.968 |
| | DivEye (Ours) | 80.52% | 0.951 |
| | Binoculars [46] | 69.32% | - |
| | RADAR [49] | 63.9% | 0.819 |
| | GLTR [39] | 51.5% | 0.709 |

cause `DivEye` only requires a single forward pass through a small GPT-2 model and performs lightweight statistical computations, it is significantly faster and more resource-efficient than larger fine-tuned transformers. This enables deployment in latency-sensitive environments without compromising performance.

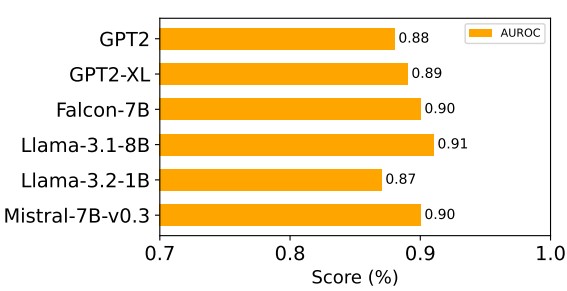

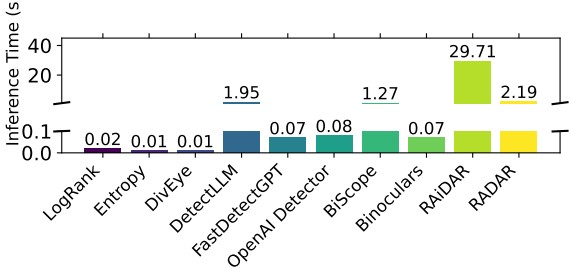

(a) `DivEye` using diverse base models.

(b) Inference time comparison.

Figure 6: (a) Performance of `DivEye` across different base models (GPT-2, GPT-2-XL, Falcon-7B). (b) Inference time (in sec) comparison of various methods.

## 4.5  Effectiveness of Boosting by `DivEye`

We empirically verify that `DivEye`-based diversity features can act as performance boosters for a wide range of detection models. To integrate `DivEye`, we concatenate its feature vector with the original model's prediction scores and train a lightweight XGBoost classifier as a meta-model.

Table 4 illustrates improvements in AUROC and AvgAcc when diversity metrics are appended to existing frameworks such as RADAR, Binoculars, DetectLLM, BiScope and FastDetectGPT. Across all evaluated baselines, the inclusion of diversity features consistently leads to better detection scores by over 18.7%. Additionally, existing frameworks in combination with `DivEye` demonstrate substantial performance gains when evaluated against paraphrasing attacks. This validates the hypothesis that static and dynamic surprisal-based features capture orthogonal information to traditional heuristics, making them a valuable addition to any detection pipeline. We also investigate how much `DivEye` contributes to the final prediction relative to the underlying detector. Appendix D.4 reports a detailed feature-importance analysis, highlighting the complementary signals captured by `DivEye` within the boosted ensembles.

## 4.6  Ablation Studies

**`DivEye`'s Performance on Different Base Models.**   To assess the adaptability of `DivEye` across different language model backbones, we evaluate its performance, on Testbed 4 of the MAGE benchmark, when instantiated with various base LLMs used to compute token-level surprisal. As shown in Figure 6a, `DivEye` consistently performs well across all models, achieving an AUROC of 0.88 with `GPT2`, 0.91 with `Llama-3.1-8B`, and 0.90 with `Falcon-7B`. Notably, even the smallest model, `GPT2`, performs competitively, and human classification accuracy improves with larger models, suggesting that higher-capacity LMs better capture stylistic diversity. These results highlight `DivEye`'s robustness and efficiency across scales, making it suitable for resource-constrained settings. Appendix D.3 further reports baseline and `DivEye` performance across various base models.

**Relevance of `DivEye`'s Features.**   `DivEye`'s feature set (Equation 6) captures token-level surprisal patterns across multiple orders, including distributional moments, first-order shifts, and second-order dynamics. To assess the contribution of each group, we compute feature importances from a trained XGBoost model. On average, second-order features contribute the most (39.4%), followed by distributional features (34.2%) and first-order differences (23.7%). The prominence of second-order features suggests that abrupt or unnatural shifts in predictability are strong indicators of machine-generated text. While traditional distributional statistics remain informative, they are insufficient on their own. These findings support `DivEye`'s central claim about second-order features: modeling the evolution of surprisal yields stronger detection capabilities than relying solely on static measures. Additionally, Appendix D.6 contains a detailed discussion about the contribution of each component in `DivEye`, including both feature-importance estimates and statistical leave-one-out ablation results.

## 5 Conclusion

We introduce `DivEye`, a lightweight classifier for AI-text detection that leverages zero-shot diversity features from token-level surprisal. Our method is model-agnostic, computationally efficient, and demonstrates strong generalization across detectors and datasets. Appendix J provides a detailed discussion of the method's limitations, broader impacts, and associated ethical considerations.

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

# A   More Details on Related Work

In recent years, the challenge of identifying AI-generated text has garnered significant attention, giving rise to a variety of detection approaches. These methods largely fall into two categories: watermark-based techniques and zero-resource detection.

**Watermarking.**   Watermarking embeds traceable patterns in a model's outputs during training or generation, enabling downstream identification of machine-generated content (Ren et al., 2024; Liu et al., 2024a). While watermarking can be effective in controlled environments, it relies on access to or cooperation from the model's developers, an assumption that frequently fails in real-world or adversarial scenarios. Furthermore, it is inherently unsuitable for practical situations where AI-generated text lacks any embedded watermark. This limitation has led to growing interest in zero-resource detection methods, which make no assumptions about access to the model's internals or training data. Instead, these methods analyze the output text alone, offering a more flexible and broadly applicable approach. Within this space, techniques can be further categorized into fine-tuned methods, which rely on labeled datasets, and zero-shot methods, which generalize to unseen models without task-specific training.

**Fine-tuned Detection.**   Fine-tuned detection methods represent a major strand of zero-resource detection, often leveraging fine-tuned classifiers built atop pre-trained language models (PLMs). A pivotal development was the Grover model, which demonstrated that models trained on text from specific generators can achieve high accuracy on in-distribution data, particularly when integrating Grover-specific layers. This inspired a wave of PLM-based detectors, most notably OpenAI's GPT-2 detector (Solaiman et al., 2019), which uses a RoBERTa classifier trained on GPT-2 outputs. However, such detectors often struggle to generalize across models, especially as newer LLMs introduce more fluent and coherent outputs.

To improve generalization and robustness, recent work has focused on feature augmentation. Stylometric approaches, for instance, introduce handcrafted features that capture writing style discrepancies between humans and machines (Mikros et al., 2023). These include measures of phraseology, punctuation, linguistic diversity, and journalistic standards, which have proven useful for detecting AI-generated tweets and news articles. Additional features such as perplexity statistics, sentiment, and error-based cues like grammatical mistakes further enrich detection pipelines (Kumarage et al., 2023).

Parallel efforts have explored structural features, incorporating models that explicitly account for the factual or contextual structure of text. Techniques such as TriFuseNet combine stylistic and contextual branches with fine-tuned BERT models, while others employ attentive-BiLSTMs to replace standard feedforward layers, enhancing interpretability and robustness (Liu et al., 2024b).

Despite these advancements, fine-tuned detectors still require labeled training data and model-specific tuning of PLMs, which can limit their scalability to novel or proprietary LLMs. Although these detectors perform exceptionally well on data similar to their training sets, they face significant drawbacks, most notably, a tendency to overfit to specific domains and a reliance on retraining for every newly emerging AI model, which is unsustainable in light of the fast-paced evolution of generative technologies. This motivates the development of methods, that leverage zero-shot features, such as `DivEye`, that aim to detect AI-generated text without relying on supervised learning or access to model internals.

**Zero-shot Detection.**   Recent research has focused on zero-shot detection strategies that require no fine-tuning on labeled examples from the target generator. These methods typically leverage statistical cues from PLM's output distributions or repurpose LLMs themselves as detectors.

A prominent class of zero-shot detectors exploits the probability structure of text under language models. DetectGPT (Mitchell et al., 2023) detects machine-generated text by measuring how strongly the log-likelihood drops under small semantic perturbations, leveraging the hypothesis that AI text lies in regions of higher negative curvature than human text. On the other hand, FastDetectGPT (Bao et al., 2024) eliminates the need for explicit perturbations by directly measuring curvature in conditional probabilities, observing that AI text typically exhibits sharper transitions between tokens compared to human writing. These observations are refined in DetectLLM (Su et al., 2023), which introduces the Log-Likelihood Log-Rank Ratio (LRR) and

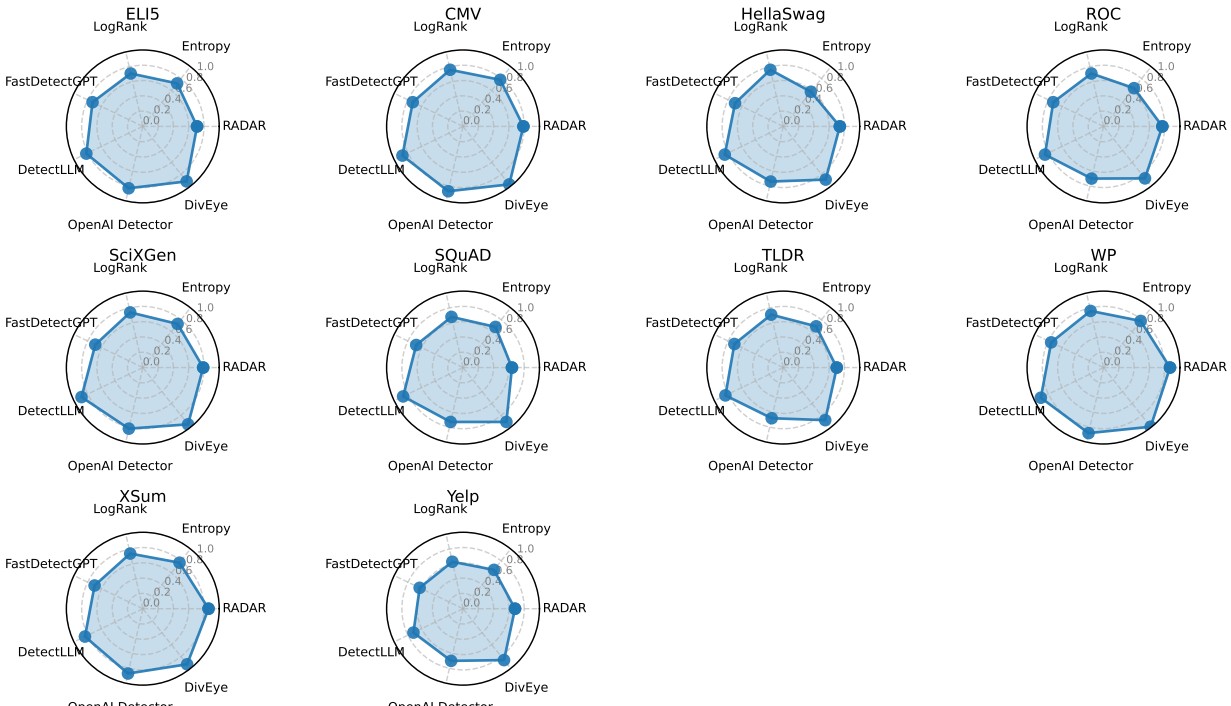

Figure 7: AUROC performance profiles of seven AI-detection tools evaluated on text generated by ten diverse domains generated by arbitrary LLMs. Each spider plot corresponds to a specific domain, with radial axes representing the AUROC score (ranging from 0 to 1) and angular axes representing the detection tools: RADAR, Entropy, LogRank, FastDetectGPT, DetectLLM, OpenAI Detector, and `DivEye`.

Normalized Perturbed log-Rank (NPR) metrics to quantify the distinguishability of AI-generated content using statistical features derived from token rankings.

Another line of work focuses on token predictability and entropy. LogRank (Ghosal et al., 2023) investigates the use of token rank distributions and demonstrates that log-rank statistics, such as the frequency of top-ranked tokens, are reliable signals of AI authorship. This builds on early work such as entropy-based detection (Lavergne et al., 2008) and GLTR (Gehrmann et al., 2019), which showed that humans tend to use more surprising and diverse tokens, while LLMs often fall back on high-probability continuations.

Moving beyond single-directional statistics, BiScope (Guo et al., 2024) proposes a bi-directional cross-entropy framework that measures how well a model's predicted logits align both with the ground truth next token (forward loss) and with the previous token (backward loss). The key insight is that AI-generated text often exhibits predictable forward progression but weaker backward association due to its autoregressive nature. A shallow classifier trained on the joint distribution of these losses can reliably detect AI text with zero-shot generalization.

Finally, Binoculars (Hans et al., 2024) offers a model-agnostic strategy by comparing the statistical disagreement between two LLMs on the same input. By contrasting the outputs of two diverse LLMs, the method detects anomalies in token distributions that are characteristic of synthetic text. This ensemble-based disagreement is found to correlate strongly with model-generated samples, providing a powerful signal without the need for training data from either model.

Collectively, these techniques demonstrate that zero-shot detection can be achieved by carefully analyzing how text aligns with the inductive biases and statistical signatures of language models, without any finetuning or access to the original generator. They lay the foundation for our proposed method, `DivEye`, which further capitalizes on diversity-based statistical properties to robustly differentiate AI- and human-written content.

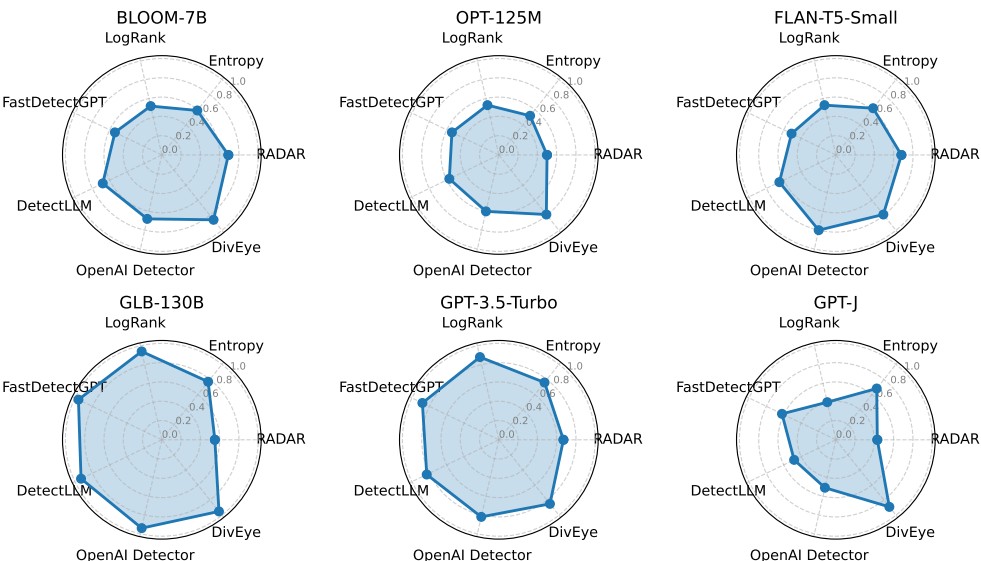

Figure 8: AUROC performance profiles of seven AI-detection tools evaluated on text generated by six different LLMs. Each spider plot corresponds to a specific language model, with radial axes representing the AUROC score (ranging from 0 to 1) and angular axes representing the detection tools: RADAR, Entropy, LogRank, FastDetectGPT, DetectLLM, OpenAI Detector, and `DivEye`.

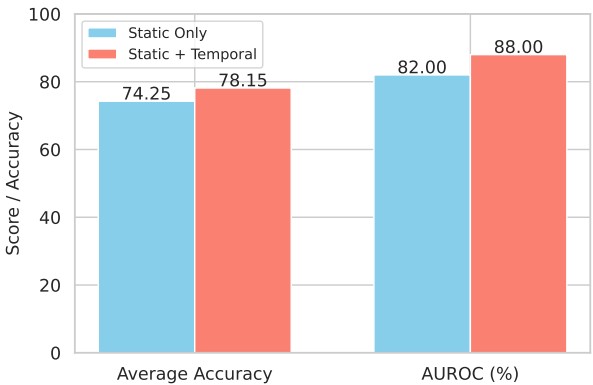

Figure 9: Ablation results on Testbed 4 of the MAGE benchmark showing the impact of temporal surprisal features. Adding temporal dynamics to static surprisal statistics improves both accuracy (from 74.25% to 78.15%) and AUROC (from 0.82 to 0.88), demonstrating their complementary value for robust AI-generated text detection.

## B  Motivation Behind Temporal Features

While static surprisal statistics such as mean, variance, skewness, and kurtosis provide useful summaries of token-level unpredictability, they overlook the evolution of this unpredictability over time, a dimension critical to distinguishing human and AI-generated text. Human authors naturally embed stylistic variability through temporal fluctuations, such as abrupt topic shifts, tonal changes, and bursts of creativity, which manifest as distinctive temporal dynamics in surprisal sequences.

Intuitively, these temporal features, as listed in Section 3, expose rhythmic and non-stationary patterns characteristic of human creativity and coherence, typically absent in the more uniform output of large language models.

Table 5: TPR@FPR=5% on Testbed 8 (Unseen Domains & Models).

|  | **LogRank** | **Entropy** | **DetectLLM** | **FastDetect** | **Binoculars** | **BiScope** | **DivEye** |
|---|---|---|---|---|---|---|---|
| **TPR** | 47.1 | 30.1 | 51.5 | 76.3 | 68.7 | 99.8 | 99.6 |

Furthermore, through an ablation study on Testbed 4 of the MAGE benchmark (Figure 9), we empirically show that augmenting static surprisal features with temporal metrics leads to a measurable improvement in classification accuracy. This highlights the complementary value of temporal dynamics in enhancing the robustness of AI-generated text detection. Moreover, an analysis of feature importance (Appendix D.6) reveals that temporal features collectively contribute more than static features, consistently ranking among the most informative signals for distinguishing between human and AI-generated text.

Overall, these findings motivate the inclusion of temporal surprisal features as integral components of our `DivEye` framework.

## C    Implementation of `DivEye`

We provide a detailed description of our `DivEye` implementation in Algorithm 1. This includes all steps from surprisal computation to feature extraction and final classification. We use an XGBoost classifier for binary classification as a preliminary choice, without extensive comparison to other classifiers, leaving exploration of alternative models for future work. For completeness and reproducibility, we include all additional implementation details, such as hyperparameter configurations, model architectures, and experimental testbeds, in Appendix I and Appendix H.

---

**Algorithm 1** `DivEye`: Algorithm for Feature Extraction & Training

---

**Require:** Text dataset $\mathcal{D} = \{(x_i, \ell_i)\}_{i=1}^N$, where $x_i$ is a text input and $\ell_i \in \{0, 1\}$ indicates whether it is human-written ($\ell_i = 1$) or machine-generated ($\ell_i = 0$)
**Require:** Pretrained auto-regressive language model $g_\phi$ (e.g., GPT-2)
**Require:** XGBoost classifier with hyperparameters $\Theta$
**Ensure:** Trained binary classifier $f_\theta$
1: Initialize an empty feature matrix $\mathcal{F} \leftarrow [\,]$
2: **for** each $(x_i, \ell_i) \in \mathcal{D}$ **do**
3:     Compute token-level log-likelihoods: $y_i \leftarrow g_\phi(x_i)$
4:     Convert to token-level surprisals: $s_i \leftarrow -y_i$
5:     Compute diversity features $\texttt{DivEye}(x_i) \in \mathbb{R}^9$ as described in Equation 6 using $s_i$
6:     Append $(\texttt{DivEye}(x_i), \ell_i)$ to $\mathcal{F}$
7: **end for**
8: Train binary classifier $f_\theta$ on feature set $\mathcal{F}$ using XGBoost with hyperparameters $\Theta$
9: **return** $f_\theta$

---

## D    Additional Results

In this section, we present additional supporting experiments that demonstrate the generalizability, robustness, and complementary strengths of `DivEye` through various ablation studies.

### D.1    Domain-Specific Performance of `DivEye`

Figure 7 presents the AUROC performance of seven detection methods evaluated across ten text domains (Testbed 3 of the MAGE benchmark). `DivEye` consistently achieves the highest AUROC scores in every domain - reaching up to 0.99 in WP, 0.97 in CMV, and 0.95 in SciXGen, outperforming other detectors by a notable margin. This highlights `DivEye`'s adaptability and robustness in capturing domain-specific writing

patterns that other methods frequently miss. These results reinforce the advantage of leveraging surprisal features for more generalizable and context-sensitive detection of AI-generated text.

## D.2  Model-Specific Performance of `DivEye`

Figure 8 compares the AUROC performance of seven detection methods across text on generated by six different large language models (Testbed 5 of the MAGE benchmark). `DivEye` achieves the highest AUROC scores across all six models, demonstrating strong robustness (0.95 on GLB-130B, 0.89 on GPT-J, 0.85 on GPT-3.5-Turbo). This consistent performance highlights `DivEye`'s effectiveness in capturing temporal surprisal patterns that generalize well across different language model architectures, making it broadly applicable for reliable AI-generated text detection.

## D.3  Performance against other base models

We evaluate the robustness of all detectors across different backbone models and report results in Table 6. These backbone models include `GPT-2` (Radford et al., 2019), `GPT-2-XL` (Radford et al., 2019), `Falcon-7B` (Almazrouei et al., 2023), `Llama-3.2-1B` (et al., 2024a), `Llama-3.1-8B` (et al., 2024a) and `Mistral-7B-v0.3` (Jiang et al., 2023). Competing methods such as Binoculars, BiScope, and DetectLLM show moderate variation with backbone choice, while FastDetectGPT and LogRank generally underperform. These results highlight that `DivEye` maintains strong and stable detection capability regardless of the underlying base model.

Table 6: Performance of different detectors across backbone models.

| Backbone Model | DivEye | Binoculars | BiScope | LogRank | DetectLLM | FastDetectGPT |
|---|---|---|---|---|---|---|
| GPT-2 | 0.88 | 0.71 | 0.86 | 0.68 | 0.75 | 0.69 |
| GPT-2-XL | 0.89 | 0.73 | 0.86 | 0.68 | 0.76 | 0.70 |
| Falcon-7B | 0.90 | 0.73 | 0.89 | 0.70 | 0.80 | 0.72 |
| Llama-3.2-1B | 0.87 | 0.71 | 0.87 | 0.70 | 0.76 | 0.71 |
| Llama-3.1-8B | 0.91 | 0.77 | 0.90 | 0.72 | 0.81 | 0.73 |
| Mistral-7B-v0.3 | 0.90 | 0.76 | 0.90 | 0.71 | 0.80 | 0.72 |

## D.4  Relative Importance of `DivEye` in a Boosted Model

Figure 10 illustrates the relative feature importance of `DivEye` when integrated into boosted ensembles with five existing AI detectors: BiScope (Guo et al., 2024), OpenAI Detector (Solaiman et al., 2019), RADAR (Hu et al., 2023), DetectLLM (Su et al., 2023), and Binoculars (Hans et al., 2024). `DivEye` contributes significantly to the overall model, with particularly high importance when combined with RADAR (91.92%), OpenAI Detector (90.26%), and Binoculars (89.71%). Even in ensembles with more advanced detectors like BiScope, `DivEye` still adds valuable signal (32.93%). These results affirm the standalone strength of `DivEye` and its utility in hybrid detection frameworks.

## D.5  Results with Different Proprietary LLMs

Table 7 reports AUROC scores of `DivEye` on text generated by five proprietary LLMs, Claude-3 Opus, Claude-3 Sonnet, Gemini 1.0-pro, GPT-3.5 Turbo, and GPT-4 Turbo, using data provided in the BiScope paper (Guo et al., 2024) across five domains. `DivEye` achieves consistently strong performance on the Normal dataset (e.g., 1.000 on GPT-3.5 Turbo for Essay) and remains robust under paraphrased inputs, with AUROC scores generally above 0.95. These results highlight `DivEye`'s ability to generalize across diverse generation models and domains, even under text transformations.

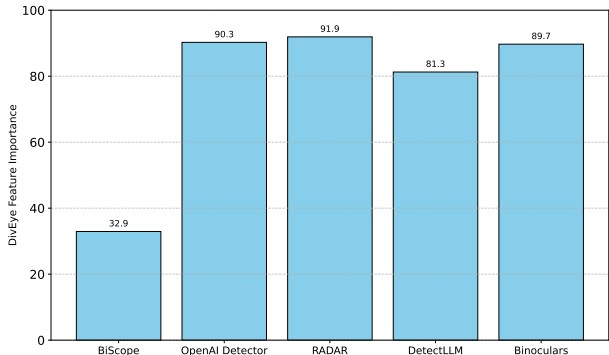

Figure 10: Feature importance of `DivEye` when integrated with various existing detectors. The plot shows how much `DivEye` contributes to the overall detection model when combined with BiScope, OpenAI Detector, RADAR, DetectLLM, and Binoculars. Higher values indicate stronger complementary impact from `DivEye`'s diversity-based features.

Table 7: AUROC scores achieved by `DivEye` on five commercial LLMs across various domains. Results are shown for both the Normal and Paraphrased datasets.

| Domain | Normal Dataset | | | | | Paraphrased Dataset | | | | |
|---|---|---|---|---|---|---|---|---|---|---|
| | Claude-3 Opus | Claude-3 Sonnet | Gemini 1.0-pro | GPT-3.5 Turbo | GPT-4 Turbo | Claude-3 Opus | Claude-3 Sonnet | Gemini 1.0-pro | GPT-3.5 Turbo | GPT-4 Turbo |
| Arxiv | 0.9942 | 0.9770 | 0.9795 | 0.9658 | 0.9793 | 0.9778 | 0.9552 | 0.9616 | 0.9689 | 0.9558 |
| Code | 0.7528 | 0.8557 | 0.7824 | 0.9577 | 0.9044 | 0.8456 | 0.9053 | 0.7521 | 0.9279 | 0.9302 |
| Creative | 0.9888 | 0.9773 | 0.9835 | 0.9951 | 0.9608 | 0.9930 | 0.9900 | 0.9957 | 0.9917 | 0.9949 |
| Essay | 0.9950 | 0.9988 | 0.9972 | 1.0000 | 0.9823 | 0.9975 | 0.9877 | 0.9814 | 0.9895 | 0.9559 |
| Yelp | 0.8855 | 0.8813 | 0.9220 | 0.8384 | 0.8942 | 0.9543 | 0.9780 | 0.9683 | 0.8524 | 0.9571 |

### D.6 Feature Importance of `DivEye`

Figure 11 presents the relative importance of each of the nine diversity-based features incorporated in `DivEye`, which are derived from surprisal statistics as detailed in Equation equation 6. The feature importances, ranging from approximately 8.1% to 13.0%, indicate that all features contribute meaningfully to model decisions, with temporal features such as, $\Delta\mu$, entropy of second derivatives $H_{\Delta^2}$, and autocorrelation $\rho_{\Delta^2}$ exhibiting the highest influence. This balanced contribution underscores the complementary nature of these statistical descriptors in enhancing `DivEye`'s detection capability when combined with existing baseline detectors.

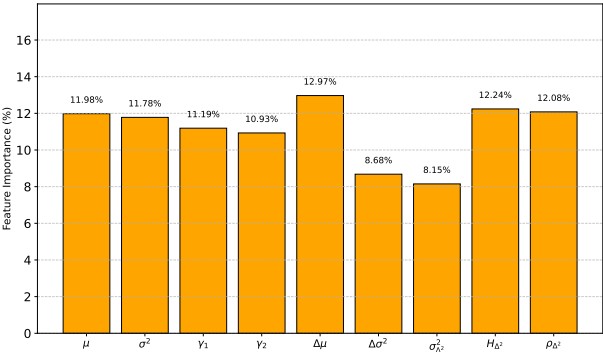

Figure 11: Relative feature importances for the nine diversity-based features used in `DivEye`. The features, as listed in Equation equation 6, represent distinct surprisal-based statistics. Higher percentages indicate greater influence in model decisions when combined with existing detectors.

### D.6.1 Statistical Relevance of `DivEye`

To evaluate the individual contribution of each component in our diversity vector $\mathcal{D}$ (Equation 6), we perform a leave-one-out ablation study. Each feature is removed individually from the 9-dimensional vector, the classifier is retrained, and the resulting performance is measured by the drop in AUC on Testbed 4 of the MAGE benchmark. To also assess statistical significance, we conduct a paired bootstrap test, resampling the test set to compute p-values under the null hypothesis that both models perform equally well. Table 8 summarizes both the AUC drop and the p-value for each feature.

Table 8: Leave-one-out feature ablation and statistical significance for `DivEye`. Each feature's removal leads to a measurable drop in AUC and is statistically significant ($p < 0.05$).

| Feature | Category | AUC Drop | p-value |
|---|---|---|---|
| $H_{\Delta^2}$ | 2nd-Order | -0.0263 | 0.001 |
| $\gamma_1$ | Distribution | -0.0239 | 0.004 |
| $\rho_{\Delta^2}$ | 2nd-Order | -0.0152 | 0.012 |
| $\sigma_s^2$ | Distribution | -0.0115 | 0.016 |
| $\Delta\sigma^2$ | 1st-Order | -0.0103 | 0.017 |
| $\gamma_2$ | Distribution | -0.0091 | 0.019 |
| $\Delta\mu$ | 1st-Order | -0.0056 | 0.027 |
| $\mu_s$ | Distribution | -0.0013 | 0.032 |
| $\sigma_{\Delta^2}$ | 2nd-Order | -0.0008 | 0.034 |

Several key insights emerge from this analysis:

- **Second-order features are the most impactful.** Removing second-order entropy $H_{\Delta^2}$ results in the largest decline in AUC, followed by the second-order autocorrelation $\rho_{\Delta^2}$, highlighting the importance of modeling higher-order dependencies in token-level surprisal dynamics.

- **Distributional features are significant.** Skewness ($\gamma_1$) and variance ($\sigma_s^2$) contribute meaningfully, indicating that asymmetry and dispersion in surprisal values enhance `DivEye`'s discriminative performance.

- **First-order features contribute consistently.** The mean and variance of first-order differences ($\Delta\mu$, $\Delta\sigma^2$) produce measurable gains, reflecting local variation in surprisal between adjacent tokens.

Even features with small absolute AUC drops, such as $\mu_s$ and $\sigma_{\Delta^2}$, are statistically significant ($p < 0.05$). This demonstrates that each feature contributes non-redundant information, supporting our core hypothesis that diversity in token-level surprisal; capturing both distributional asymmetries and temporal patterns is essential for detecting machine-generated text.

### D.7 Performance against same model

To investigate whether `DivEye`'s detection ability relies on a distributional mismatch between the generator and the surprisal model, we conducted a controlled experiment using the same model for both purposes. Specifically, we computed diversity-based features and trained the `DivEye` classifier using three different LLMs: `Falcon-7B`, `Llama-3.1-8B`, and `GPT-2-XL`.

We further evaluated generalization by performing an out-of-distribution test using generations from 500 prompts drawn from the OASST (Köpf et al., 2023) and Self-Instruct (Wang et al., 2023) datasets. Table 9 reports the resulting AI detection accuracies.

Despite using the same model for both generation and surprisal estimation, `DivEye` maintains high classification accuracy across all settings. This demonstrates that `DivEye`'s effectiveness stems from intrinsic statistical patterns in the generated text rather than artifacts arising from a model mismatch.

Table 9: `DivEye` performance when using the same model for both generation and surprisal computation.

| Model | AI Accuracy (%) |
|---|---|
| Falcon-7B | 98.2 |
| Llama-3.1-8B | 96.1 |
| GPT-2-XL | 98.8 |

### D.8 Performance against Longformer

Table 10: AUROC comparison of `DivEye`, `DivEye` (w/ BiScope), and Longformer across MAGE test settings.

| Setting | Longformer | DivEye | DivEye (w/ BiScope) |
|---|---|---|---|
| Fixed-domain, Model-specific | 0.990 | 0.994 | 1.000 |
| Arbitrary-domains, Model-specific | 0.990 | 0.972 | 0.991 |
| Fixed-domain, Arbitrary-models | 0.990 | 0.993 | 0.998 |
| Arbitrary-domains, Arbitrary-models | 0.990 | 0.880 | 0.934 |
| OOD: Unseen Models | 0.950 | 0.859 | 0.952 |
| OOD: Unseen Domains | 0.930 | 0.975 | 0.989 |
| OOD: Unseen Domains & Models | 0.940 | 0.924 | 0.986 |
| Paraphrasing Attacks | 0.750 | 0.870 | 0.923 |

Longformer, being a fine-tuned detector, was not included in Table 1 since its setup differs fundamentally. Nonetheless, for completeness, we provide a detailed AUROC-based comparison of `DivEye` against Longformer across the 8 challenging MAGE testbeds.

As shown in Table 10, `DivEye` consistently matches or outperforms Longformer in most settings. Even under OOD scenarios and paraphrasing attacks, `DivEye` demonstrates strong generalization, often exceeding Longformer's performance. `DivEye` (w/ BiScope) further improves AUROC across nearly all testbeds, highlighting the benefits of incorporating diverse zero-shot features.

### D.9 Performance across edge-cases

### D.9.1 Performance across non-native-written texts

To evaluate potential bias against non-native speakers, a known vulnerability in many perplexity-based detectors, we conducted a targeted evaluation using the COREFL dataset (Lozano et al., 2020). This dataset consists of 1426 essays written by native German and Spanish speakers, explicitly categorized by their English proficiency levels (ranging from A1 to C2).

We trained (or configured) `DivEye`, Binoculars and FastDetectGPT using a balanced set comprising 75% of the COREFL human texts and an equivalent number of AI-generated samples drawn from MAGE Testbed 4. We evaluate performance on the held-out 25% to measure detection accuracy across proficiency brackets.

Table 11: Detection Accuracy on Non-Native English Writing.

| Framework | Average Accuracy |
|---|---|
| Binoculars | 68.34% |
| FastDetectGPT | 71.71% |
| **DivEye (Ours)** | **82.07%** |

As shown in Table 11, `DivEye` significantly outperforms existing baselines in identifying non-native human text, achieving a robust 82.1% average accuracy compared to 68.34% for Binoculars and 71.7% for FastDetectGPT.

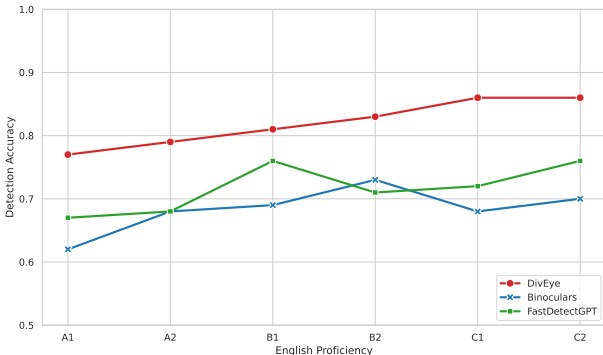

Figure 12: Granular depiction of various frameworks performance on various non-native texts.

Crucially, as illustrated in Figure 12, our granular analysis reveals that performance of all frameworks correlates with linguistic complexity: A1 texts (the most beginner class) are the most difficult to detect, while accuracy steadily improves as proficiency rises. As for `DivEye`, this suggests that detection becomes easier as the text exhibits the richer rhythmic signatures of advanced writing.

### D.9.2 Performance across legal & professional documents

A common criticism in AI text detectors is their potential fragility when applied to highly formal human writing, such as legal contracts or technical specifications. In these domains, human writers often strive for low entropy and rigid structure, traits typically associated with machine generation. To address this concern, we evaluated `DivEye` on the LEDGAR dataset (Tuggener et al., 2020), a corpus of legal provisions extracted from US SEC filings.

Table 12: Detection Accuracy on LEDGAR dataset.

| Framework | Accuracy |
|---|---|
| Binoculars | 91.67% |
| FastDetectGPT | 94.8% |
| **DivEye (Ours)** | **99.49%** |

We adopted a robust evaluation protocol using a 75-25 train-test split to ensure generalization. As shown in Table 12, despite the constrained and repetitive nature of contract language, `DivEye` successfully outperforms other frameworks on this task. These findings provide empirical evidence that `DivEye`'s core hypothesis holds even in challenging edge cases.

### D.9.3 Visualizing texts under observation

Analyzing the probability distributions across domains reveals a distinct hierarchy of detectability. We have already seen a distribution in Figure 3 for `DivEye`'s effectiveness on MAGE Testbed 4.

Contrary to the concern that formal writing might resemble algorithmic generation, we find that in Figure 13 professional legal text (LEDGAR) is the most easily distinguishable from AI, with its probability distribution shifted furthest towards the human extreme.

This suggests that while professional writing is structured, it contains sharp, domain-specific peaks in information density that are representative of human writing. Furthermore, Standard (MAGE Testbed 8) and Non-Native (COREFL) texts exhibit remarkably similar distributions, effectively overlapping in the high-probability region. We hope to investigate this further in our future works.

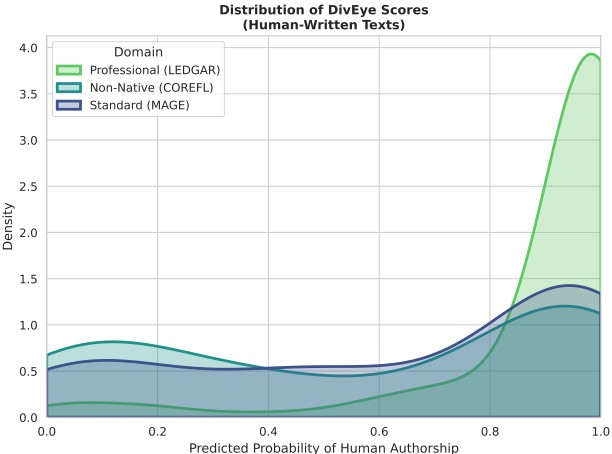

Figure 13: Predicted distributions of predicted class probabilities for diverse human written texts. We utilize the LEDGAR, COREFL and the MAGE Testbed 8 texts to plot this distribution.

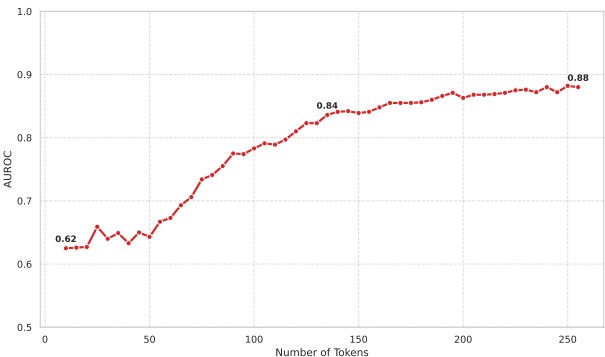

Figure 14: `DivEye`'s performance on different text lengths (characterized by tokens) on MAGE Testbed 4.

### D.9.4 Text sensitivity analysis of `DivEye`

To determine the minimum linguistic context required for reliable detection, we evaluated the performance of `DivEye` across varying text lengths. We truncated test samples to specific token counts ranging from $N = 10$ to $N = 256$ and measured the AUROC at each interval. As illustrated in Figure 14, performance follows a predictable scaling law relative to information availability.

Detection is constrained on extremely short fragments, yielding an AUROC of 0.62 at 10 tokens; this is expected, however, the system demonstrates rapid convergence: performance improves steadily as the context window expands, reaching a robust AUROC of 0.88 by 256 tokens. This confirms that while `DivEye` benefits from longer contexts, it effectively captures the structural signature of human writing within standard paragraph lengths.

We discuss short length texts as a limitation in Appendix J.

## E   Additional Adversarial Attacks on `DivEye`

We evaluate `DivEye` under a range of adversarial settings, including character-level perturbations, word- and phrase-level edits, paraphrasing, prompt obfuscations, and distribution shifts (temperature changes and degenerate sampling), to comprehensively assess its robustness.

Table 13: Performance of `DivEye` and open-source baselines on all listed adversarial attacks on the RAID benchmark.

| Settings | Methods | FPR@TPR=5% |
|---|---|---|
| **[RAID] Adversarial Attacks** | | |
| Whitespace Attack | Desklib AI | 94.9% |
| | e5-small-lora | 93.9% |
| | DivEye (Ours) | 79.8% |
| | Binoculars | 68.7% |
| | RADAR | 61.1% |
| | GLTR | 43.1% |
| Upper-Lower Attack | Desklib AI | 87.2% |
| | e5-small-lora | 93.9% |
| | DivEye (Ours) | 85.3% |
| | Binoculars | 72.8% |
| | RADAR | 65.1% |
| | GLTR | 45.3% |
| Synonym Attack | Desklib AI | 80.6% |
| | e5-small-lora | 85.6% |
| | DivEye (Ours) | 67.1% |
| | Binoculars | 42.1% |
| | RADAR | 62.7% |
| | GLTR | 28.7% |
| Paraphrase Attack | Desklib AI | 83.7% |
| | e5-small-lora | 85.5% |
| | DivEye (Ours) | 74.4% |
| | Binoculars | N/A |
| | RADAR | 62.4% |
| | GLTR | 43.0% |
| Perplexity Misspelling | Desklib AI | 92.9% |
| | e5-small-lora | 92.5% |
| | DivEye (Ours) | 90.6% |
| | Binoculars | 77.2% |
| | RADAR | 64.3% |
| | GLTR | 57.0% |

| Settings | Methods | FPR@TPR=5% |
|---|---|---|
| Number Attack | Desklib AI | 93.0% |
| | e5-small-lora | 93.5% |
| | DivEye (Ours) | 92.1% |
| | Binoculars | 76.4% |
| | RADAR | 65.7% |
| | GLTR | 57.3% |
| Insert Paragraph | Desklib AI | 94.9% |
| | e5-small-lora | 93.9% |
| | DivEye (Ours) | 92.2% |
| | Binoculars | 70.7% |
| | RADAR | 68.2% |
| | GLTR | 58.3% |
| Homoglyph Attack | Desklib AI | 99.7% |
| | e5-small-lora | 11.1% |
| | DivEye (Ours) | 61.6% |
| | Binoculars | 36.1% |
| | RADAR | 44.8% |
| | GLTR | 20.3% |
| Article Deletion | Desklib AI | 90.5% |
| | e5-small-lora | 92.0% |
| | DivEye (Ours) | 88.0% |
| | Binoculars | 73.3% |
| | RADAR | 63.0% |
| | GLTR | 48.9% |
| Alt. Spelling Attack | Desklib AI | 94.3% |
| | e5-small-lora | 93.4% |
| | DivEye (Ours) | 92.01% |
| | Binoculars | 77.6% |
| | RADAR | 65.5% |
| | GLTR | 58.2% |
| Zero Width Space | Desklib AI | 87.5% |
| | e5-small-lora | 93.9% |
| | DivEye (Ours) | 92.0% |
| | Binoculars | 98.4% |
| | RADAR | 78.4% |
| | GLTR | 97.9% |

## E.1 Adversarial Attack Analysis of `DivEye`

We evaluate `DivEye` against a wide range of adversarial attacks using the RAID benchmark, reporting average classification accuracies across all attack categories listed in Table 13. `DivEye` achieves performance on par with the top-performing fine-tuned models reported by the benchmark. Notably, it consistently surpasses all zero-shot detectors by a significant margin across every attack type, demonstrating strong robustness against both diverse adversarial attacks.

## E.2 Detection against other diverse online paraphrasers

To evaluate the robustness of `DivEye` against paraphrasing attacks intended to "humanize" AI-generated text, we curate a set of 21 arXiv abstracts generated by Claude-3.5-Sonnet and paraphrase each using three widely used commercial tools: ZeroGPT[2], GPTinf[3], and QuillBot[4]. This results in 63 paraphrased texts (21 per tool), each aiming to evade AI detectors through stylistic and lexical variation. We provide this smaller dataset in the supplementary materials and in our anonymous repository.

---

[2]https://www.zerogpt.com/

[3]https://app.gptinf.com/

[4]https://quillbot.com/paraphrasing-tool

Table 14: Detection performance of `DivEye` against paraphrased outputs generated by three commercial tools. Each model was tested on 21 samples per paraphraser.

| Paraphraser | MAGE Testbed 4 Model | Claude-3.5-Sonnet Model |
|---|---|---|
| Claude-3.5-Sonnet (original) | 20 / 21 | 21 / 21 |
| GPTinf | 18 / 21 | 19 / 21 |
| ZeroGPT | 20 / 21 | 17 / 21 |
| QuillBot | 20 / 21 | 17 / 21 |

We assess detection performance using two XGBoost classifiers trained exclusively on `DivEye` features: one trained on MAGE's Testbed 4 (Arbitrary Models & Arbitrary Domains), and another trained on 280 Claude-3.5-Sonnet generated arXiv abstracts (from BiScope (Guo et al., 2024)). The results, presented in Table 14, highlight `DivEye`'s ability to maintain detection accuracy even in the presence of strong paraphrasing transformations.

### E.3 Dependence of `DivEye` on Generation Temperature

We investigate the influence of generation temperature on `DivEye`'s detection performance to evaluate its robustness against variations in text predictability. Specifically, we conduct two experiments on the MAGE benchmark: intra-model temperature variation and cross-model variable-temperature detection.

#### E.3.1 Intra-Model Temperature Variation

Using `GPT-2` as the zero-shot feature generator, we evaluate `DivEye` across a wide range of sampling temperatures (default $T = 1.0$). AUROC results for selected MAGE testbeds are presented in Table 15.

Table 15: `DivEye` AUROC across different temperatures for GPT-2 generated texts.

| Testbeds / Temperatures | $T = 0.1$ | $T = 0.3$ | $T = 0.5$ | $T = 0.7$ | $T = 1.0$ | $T = 1.2$ | $T = 1.4$ | $T = 1.6$ |
|---|---|---|---|---|---|---|---|---|
| Arbitrary Domains & Arbitrary Models | 0.8784 | 0.8760 | 0.8776 | 0.8886 | 0.8825 | 0.8767 | 0.8842 | 0.8698 |
| Unseen Models (GPT-3.5-Turbo, OOD) | 0.8473 | 0.8432 | 0.8595 | 0.8619 | 0.8617 | 0.8583 | 0.8488 | 0.8567 |

Results indicate that `DivEye`'s AUROC remains consistently high across all temperatures. Even at extreme sampling regimes, performance does not degrade, suggesting `DivEye` captures stable distributional signals across different entropy levels within the same generator.

#### E.3.2 Cross-Model, Variable-Temperature Detection

We further test `DivEye` on `Llama-3.1-8B`, generating 50 samples per temperature (ranging $T = 0.1$ to 1.6) using OASST prompts. This simulates an adversarial generator varying sampling temperature to evade detection. Table 16 summarizes AI detection accuracy.

Table 16: `DivEye` AI detection accuracy for `Llama-3.1-8B` across different temperatures.

| Temperature | AI Accuracy (%) |
|---|---|
| $T = 0.1$ | 94.0 |
| $T = 0.3$ | 96.0 |
| $T = 0.5$ | 100.0 |
| $T = 0.7$ | 96.0 |
| $T = 1.0$ | 96.0 |
| $T = 1.2$ | 98.0 |
| $T = 1.4$ | 94.0 |
| $T = 1.6$ | 96.0 |

These experiments demonstrate that `DivEye` maintains strong performance across both intra-model and cross-model temperature variations, consistently achieving high AUROC and detection accuracy. Even at high temperatures ($T = 1.6$), where generations are more diverse, `DivEye` remains robust, highlighting its resilience against temperature-based evasion strategies in real-world deployment.

### E.4 Robustness of `DivEye` to low-quality LMs and Prompt-based attacks

We further evaluate `DivEye`'s robustness against two challenging conditions raised by the reviewer: degenerate or less predictable generators, and prompt-level adversarial obfuscations.

#### E.4.1 Performance on Less Predictable Generators

While `DivEye`'s primary focus is detecting outputs from realistic, high-quality LLMs, we also assess its behavior on weaker or degenerate text sources to explore the method's boundaries. The RAID benchmark already includes a variety of degenerate and obfuscation-style perturbations - such as synonym replacement, paraphrasing, number swaps, homoglyph substitutions, and zero-width spaces - which `DivEye` handles effectively (see Table 13 for results).

To complement these benchmark results, we evaluate two baseline degenerate generators producing 500 samples each:

1. **Random Token Generator**: uniformly samples tokens from `GPT-2`'s vocabulary to generate incoherent sequences without semantic structure.

2. **Keyword-Stuffing Generator**: repeats high-frequency topical keywords in ungrammatical, repetitive patterns.

Using `GPT-2` and Testbed 4 of the MAGE benchmark, `DivEye` achieves near-perfect AI detection: 99.99% on the random token set and 99.95% on the keyword-stuffed set. These results indicate that `DivEye` confidently flags incoherent or low-quality text as non-human-written, suggesting that it is sensitive to general non-human-likeness rather than relying solely on repetition or frequency patterns.

#### E.4.2 Robustness to Prompt-Based Adversarial Attacks

We also assess `DivEye` under adversarial prompt-based obfuscation, designed to disrupt `GPT-2`'s predictability patterns. Specifically, we generated 250 texts from `GPT-4o` and `Llama-3.1-8B` using the following system prompt to encourage unusual and unpredictable writing styles:

> *"You are a highly creative and unconventional writer. Your goal is to respond with original, imaginative, and surprising ideas that avoid clichés, common phrases, or predictable completions. Do not repeat yourself or follow standard templates. Instead, embrace novelty, abstract connections, and lateral thinking. Unusual metaphors, unexpected analogies, and rare vocabulary are encouraged."*

These generations were sampled from different models and prompts, constituting a fully out-of-distribution evaluation. Table 17 shows `DivEye`'s detection performance compared to Binoculars.

Table 17: AI detection accuracy under prompt-based adversarial attacks.

| Model | DivEye Accuracy (%) | Binoculars Accuracy (%) |
|---|---|---|
| Llama-3.1-8B | 90.8 | 72.4 |
| GPT-4o | 92.4 | 78.0 |

Despite these obfuscation strategies, `DivEye` maintains high accuracy, substantially outperforming Binoculars. This demonstrates that `DivEye`'s diversity-based signals remain robust under both low-quality generators and adversarial prompting, highlighting its resilience in real-world deployment scenarios.

Overall, these experiments confirm that `DivEye` effectively detects both degenerate text and highly unpredictable LLM outputs, providing strong evidence of its general robustness.

### E.5 Dependence of `DivEye` on Top-$p$ Generations

To ensure `DivEye`'s robustness extends beyond temperature scaling, we evaluated its stability under varying nucleus sampling (top-$p$) thresholds. We conducted an evaluation using `Llama-3.1-8B`, generating 100 text samples per threshold across a broad spectrum of values $p \, in\{0.1, 0.3, 0.5, 0.8, 0.95\}$. The detection results are summarized in Table 18.

Table 18: Detection Accuracy of `DivEye` on `Llama-3.1-8B` across varying Top-$p$ thresholds.

| Top-$p$ Threshold | AI |
|---|---|
| $p = 0.1$ | 96% |
| $p = 0.3$ | 94% |
| $p = 0.5$ | 97% |
| $p = 0.8$ | 93% |
| $p = 0.95$ | 95% |
| **Average** | **95%** |

As evidenced by the results, `DivEye` demonstrates remarkable consistency across the entire range of $p$ values, with an average accuracy of 95% and negligible variance. This stability can be attributed to the nature of the sampling mechanism: while top-p removes low-probability tokens and may affect the absolute mean perplexity, it fails to emulate the unpredictability characteristic of human writing.

## F  Additional Discussions

### F.1 Robustness of Binoculars under OOD Conditions

While Binoculars achieves high performance in its original paper, with AUROC values consistently above 0.99 (Tables 3 and 4), its robustness under out-of-distribution (OOD) conditions is substantially weaker. For example, Tufts et al. (2025) (Table 13) shows a noticeable drop in AUROC when Binoculars is applied to datasets with distributions different from its training set, despite remaining competitive. Similarly, in the Voight-Kampff Generative AI Authorship Verification Challenge 2024 (Ayele et al., 2024), Binoculars underperformed significantly under highly OOD conditions, failing to replicate its originally reported accuracy and AUROC.

These findings are consistent with our observed AvgAcc of 79%, reflecting Binoculars' sensitivity to domain shift. Importantly, our evaluation deliberately includes diverse scenarios to rigorously test generalization beyond training conditions. This approach better reflects real-world deployment, where robustness to OOD tasks is critical. Our results do not contradict prior work; rather, they reinforce the understanding of Binoculars' limitations under distribution shifts.

### F.2 Practical constraints on adversarial attacks against `DivEye`

The possibility of targeted attacks that attempt to evade `DivEye` by steering autoregressive models to generate tokens falling in low-probability regions under `GPT-2`'s distribution seems like a plausible idea to evade detection. While such attacks are theoretically conceivable, implementing them in practice is extremely challenging.

Autoregressive models do not natively support fine-grained constraints that enforce divergence from another model's token-level distribution without compromising fluency or coherence. Even for large models such as `GPT-4o`, generating text that is simultaneously plausible and systematically unpredictable to a specific detector like `DivEye` is highly nontrivial.

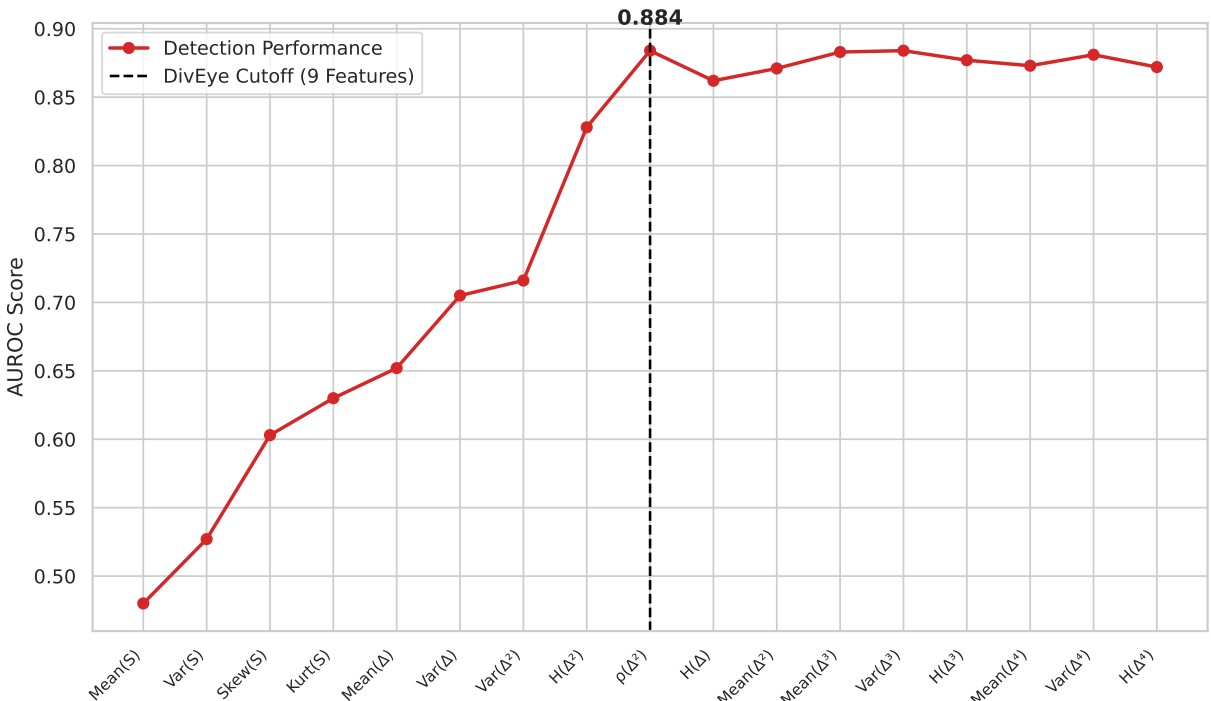

Figure 15: Inclusion of additional features into `DivEye` results in diminishing returns in AUROC. This experiment was run on MAGE Testbed 4.

Furthermore, `DivEye` is designed to generalize across diverse model backbones. Our best-performing variant, for example, uses `Llama-3.1-8B` as the surprisal scorer, which has a larger vocabulary and greater expressive capability than GPT-2. This substantially increases the difficulty for an adversary to generate text that appears unpredictable to the detector while remaining coherent and human-like.

Taken together, these considerations suggest that although targeted, detector-specific attacks are theoretically possible, they are rare and practically hard to execute in realistic generation pipelines.

### F.3 Diminishing returns in expanding `DivEye`

`DivEye` was deliberately designed with a concise set of nine features to balance computational efficiency with interpretability. While the architecture is flexible and allows researchers to integrate additional signals, our empirical analysis suggests that the current feature set captures the vast majority of the discriminative signal. To validate this design choice, we conducted a scaling law experiment evaluating the marginal utility of incorporating higher-order surprisal dynamics.

We extended the original feature vector by computing third-order ($\Delta^3$) and fourth-order ($\Delta^4$) derivatives of the surprisal sequence, extracting the mean, variance, and entropy for each. We also include missing features, including the mean of the second-order and entropy of the first-order of the surprisal sequence. This resulted in an expanded candidate set of 17 features. We then trained a series of XGBoost classifiers, iteratively adding one feature at a time in hierarchical order and measuring the resulting AUROC on MAGE Testbed 4.

As illustrated in Figure 15, the detection performance improves sharply with the inclusion of the first-order and second-order `DivEye` features, effectively plateauing at the ninth feature (second-order autocorrelation). The addition of third- and fourth-order features yields negligible gains, confirming a law of diminishing returns.

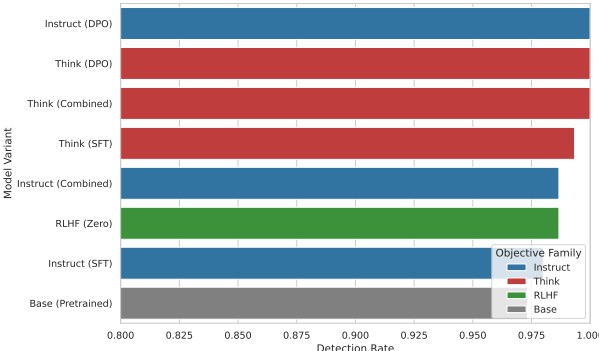

Figure 16: Impact of post-training objectives on detectability; `DivEye` consistently maintains or slightly enhances detectability compared to the unaligned Base model.

### F.4 Robustness across post-training objectives

Modern LLMs undergo extensive post-training alignment to enhance instruction following and safety. A critical question is whether these alignment steps alter the statistical surprisal signature of the generated text, potentially degrading detection performance compared to the pre-trained base model.

To investigate this, we conducted a controlled experiment using the `OLMo-3-7B` (Olmo et al., 2025) model family, which provides open access to checkpoints at every training stage. We curated a dataset (of 600 texts from diverse domains, different temperatures and top-$p$) comprising eight distinct model variants, including the unaligned Base model, Instruct-tuned versions, Reasoning models, and RLVR baselines. We adopted a unified training strategy: a single `DivEye` detector was trained on an aggregate containing 75% of the data from all variants (balanced with human text), and subsequently evaluated on the held-out 25% of each specific variant.

As detailed in Figure 16, `DivEye` demonstrates remarkable stability across all training objectives. While one might hypothesize that alignment (which constrains the output space) would significantly shift the surprisal distribution, our results show that the core detectability remains high.

Contrary to the intuition that aligned models might better mimic human stylistic nuances, we observe that alignment consistently enhances detectability. The pre-trained Base model yields the lowest (yet still robust) accuracy of 97.33%, whereas models fine-tuned with DPO and Thinking capabilities achieve perfect detection rates.

This trend suggests that post-training interventions constrain the model's output distribution to a highly structured subspace. This constraint likely dampens the natural entropic variance present in the pre-training data, making it easier for `DivEye` to distinguish post-trained models from genuine human writing.

## G Failure of other motivational methods: LookForward

A natural hypothesis we considered was that LLMs, being autoregressive in nature, lack global sentence-level planning due to their left-to-right generation paradigm. Unlike humans who often write with a sense of the sentence's future, autoregressive models generate one token at a time conditioned only on the preceding context. Based on this, we hypothesized that detection features relying on this "lack of foresight" could effectively identify machine-generated text.

This suggests that the model never observes $x_{>t}$ when predicting $x_t$, whereas human writing may implicitly reflect awareness of future tokens. Our idea was to define a LookForward discrepancy by comparing model likelihoods under forward-only conditioning vs. bidirectional context.

However, our empirical evaluations demonstrate that this feature is ineffective, achieving 0.50 AUROC on diverse testbeds. As LLMs undergo extensive training and optimization, they appear to develop strong

internal planning capabilities, even in an autoregressive setting. Despite the absence of access to future tokens during generation, LLMs approximate global coherence and structure remarkably well. This aligns with recent literature suggesting that transformers internalize hierarchical and global sentence structure across layers, even when trained autoregressively.

While this method is theoretically appealing, its failure in practice highlights the difficulty of quantifying planning behavior in black-box LLMs. We hope this limitation can be better understood in the future through more fine-grained interpretability analyses of autoregressive models, which may reveal how planning and coherence emerge despite the lack of explicit future context.

## H Testbed Details

We evaluate `DivEye` on a comprehensive testbed spanning three major AI-text detection benchmarks, MAGE (Li et al., 2024), HC3 (Guo et al., 2023) & RAID (Dugan et al., 2024), covering a diverse range of domains, language models, and adversarial attacks. These benchmarks allow us to assess the generalizability and robustness of our method across realistic deployment scenarios. This section provides a comprehensive overview of the testbeds used in our evaluation, including all domains, language models, and adversarial attacks featured in the MAGE and RAID benchmarks, along with relevant configuration details.

**Details about MAGE Benchmark.** The MAGE benchmark (Li et al., 2024) comprises eight diverse testbeds designed for evaluating machine-generated text detection. Testbeds 1 through 4 include standard train, validation, and test splits, while Testbeds 5 through 8 serve as out-of-distribution (OOD) datasets, evaluated using models trained on Testbed 4. Notably, Testbed 4, Arbitrary Domains & Arbitrary Models, is the most comprehensive, enabling evaluation across the full range of domains and language models listed in the MAGE paper. Detailed information regarding dataset splits and sample counts is available in the original benchmark documentation.

MAGE covers a wide array of domains, including CMV (Tan et al., 2016), Yelp (Zhang et al., 2015), XSum (Narayan et al., 2018), TLDR, ELI5 (Fan et al., 2019), WP (Fan et al., 2018), ROC (Mostafazadeh et al., 2016), HellaSwag (Zellers et al., 2019), SQuAD (Rajpurkar et al., 2016), and SciXGen (Chen et al., 2021a). The OOD domains include CNN/DailyMail (See et al., 2017), DialogSum (Chen et al., 2021b), PubMedQA (Jin et al., 2019), and IMDb (Maas et al., 2011).

MAGE also incorporates text generated from over 27 different LLMs (Brown et al., 2020; Chung et al., 2022; et al., 2023; Sanh et al., 2022; Touvron et al., 2023a; Zeng et al., 2023; Zhang et al., 2022), enabling rigorous and varied evaluations. For further implementation specifics, readers are encouraged to consult the MAGE paper.

**Details about RAID Benchmark.** The RAID benchmark (Dugan et al., 2024) comprises over 6.2 million samples, offering extensive coverage across domains, language models, sample sizes, and adversarial attacks. It provides a clear separation into training, validation, and testing splits to support rigorous evaluation. The benchmark spans a wide range of domains, including scientific abstracts (Paul & Rakshit, 2021), book summaries (Bamman & Smith, 2013), BBC News articles (Greene & Cunningham, 2006), poems (Arman, 2020), recipes (Bień et al., 2020), Reddit posts (Völske et al., 2017), movie reviews (Maas et al., 2011), Wikipedia entries (Aaditya Bhat, 2023), Python code, Czech news (Boháček et al., 2022), and German news articles (Schabus et al., 2017).

RAID employs text generated from 11 diverse LLMs (Radford et al., 2019; MosaicML, 2023; Jiang et al., 2023; Cohere, 2024; Ouyang et al., 2022; Touvron et al., 2023b; et al., 2024b), ensuring broad model representation. Additionally, it includes over 11 adversarial attack strategies (Liang et al., 2023b;a; Wolff & Wolff, 2022; Bhat & Parthasarathy, 2020; Krishna et al., 2023; Pu et al., 2023; Gagiano et al., 2021; Guerrero et al., 2022), designed to test the robustness of detectors under challenging settings. Comprehensive descriptions and detailed results of these attacks are provided in Appendix E.1, with all results reported as of April 2025. For further implementation specifics, readers are encouraged to consult the RAID paper.

**Details about HC3 Benchmark.** The HC3 benchmark (Guo et al., 2023) offers a large-scale, multilingual dataset designed to evaluate the effectiveness of detectors in distinguishing human-written text from AI-generated responses. It encompasses both English and Chinese content, covering a wide variety of domains and question types. This bilingual setup facilitates cross-linguistic performance analysis and underscores the difficulties of achieving generalization across different languages and cultural contexts. In our experiments, we adopt an 80-20 train-test split. For comprehensive dataset numbers, we refer readers to the original HC3 paper.

## I  Hyperparameter Settings

Table 19 outlines the hyperparameter configurations used for our experiments. We utilize the XGBoost classifier with standard but tuned settings to handle class imbalance and optimize detection performance. For our proposed method `DivEye`, we set the number of bins for entropy computation to 20 and truncate input sequences at a maximum length of 1024 tokens. All experiments were run on a single NVIDIA DGX A100 (40 GB), and reported results reflect the median of three runs.

Table 19: Hyperparameters used for the XGBoost Classifier and `DivEye`.

| XGBoost Hyperparameter | Value |
| --- | --- |
| `random_state` | 42 |
| `scale_pos_weight` | $(\text{len}(Y_{\text{train}}) - \sum Y_{\text{train}})/\sum Y_{\text{train}}$ |
| `max_depth` | 12 |
| `n_estimators` | 200 |
| `colsample_bytree` | 0.8 |
| `subsample` | 0.7 |
| `min_child_weight` | 5 |
| `gamma` | 1.0 |
| **DivEye Parameter** | **Value** |
| `Entropy bins` | 20 |
| `Tokenizer Max Length` | 1024 + Truncation |

## J  Limitations, Broad Impacts, Reproducibility & Ethical Considerations

**Future Work & Limitations.** While `DivEye` demonstrates strong generalization across domains and models in zero-shot settings, several limitations suggest promising directions for future work. Our approach relies on features derived from LLM token-level behavior, which may vary across model sizes, architectures, and tokenization schemes. Although our current performance is robust, it is unclear whether we are approaching an optimal limit for AI-text detection. Moreover, our diversity metrics are less effective on very short texts, where statistical patterns are inherently limited. We hope to address these challenges in future work by exploring more adaptive teacher selection strategies and improving robustness in diverse text lengths.

**Broad Impacts.** This work introduces `DivEye`, a model-agnostic, and scalable framework for detecting AI-generated text that remains robust across models, domains, and decoding strategies. By leveraging purely intrinsic statistical features, without requiring fine-tuning or access to the internals of large language models, `DivEye` is broadly applicable and easy to deploy in real-world settings. We envision this framework as a practical tool to support responsible AI usage, aiding in the detection of synthetic text across domains such as education, journalism, and online content moderation. However, we emphasize that detection results should be interpreted with care and recommend using `DivEye` as one component within a broader, multi-layered content verification pipeline.

We deliberately restrict `DivEye` to nine features, each of which is theoretically motivated and captures a distinct aspect of surprisal diversity. While additional features could be engineered, our preliminary experiments indicated diminishing returns beyond this set. This preserves interpretability, efficiency, and robustness, while still providing strong empirical performance. We also discuss certain concerns about our results and the practicality of adversarial attacks in Appendix F.

Furthermore, our finding that a lightweight model (GPT-2) can effectively serve as an observer for state-of-the-art generators has significant implications for the democratization and sustainability of AI forensics. By demonstrating that detection does not require computational parity with the generator, `DivEye` establishes a strong lower bound for efficiency. However, for production environments where maximizing detection sensitivity is paramount, we recommend leveraging more capable LMs as the feature extractor.

**Reproducibility.** We release all code and evaluation scripts to ensure full reproducibility. Detailed training, testing and hyperparameter configurations are included in Appendices H and C.

**Ethical Considerations.** As with all AI-text detectors, `DivEye` is not infallible and may produce incorrect classifications or false positives. We emphasize that detection outputs should be treated as probabilistic signals rather than definitive evidence. When used in high-stakes settings, such as academic integrity or content moderation, additional human review and validation are essential. We encourage responsible deployment of `DivEye` to support large-scale analysis, but caution against its use in critical decision-making.

## K    Illustrative cases of `DivEye` with probabilities

We provide a few representative examples for readers in Table 20, showcasing the probability scores assigned by `DivEye` to different text sources.

| Source | Text | $Probability_{\mathbf{AI}}$ (DivEye) |
|---|---|---|
| GPT-4-Turbo | For centuries, the pursuit of immortality was the ultimate quest, a beacon drawing the brilliant and the mad alike. I, Dr. Elara Mendoza, fell somewhere in between, teetering on the precipice of genius and insanity. And after countless sleepless nights, fueled by an obsession that bordered on madness, I finally did it. I unlocked the secret to immortality. In my laboratory bathed in the cold, metallic gleam of artificial light, the hum of machinery breathed life into my creation, a serum, translucent and iridescent, a potion promising eternity. As the final drop fell into the vial, a silence descended, thick with anticipation. But in this moment of triumph, a chill swept through the room, frosting over the warmth of victory... | **0.97035** |
| Claude-3-Opus | Dressing for Success: Budgeting for Interview Attire and Work Uniforms as a Medical Office Assistant - As a medical office assistant, presenting a professional image is crucial for success in both the job interview process and daily work life. Dressing appropriately demonstrates respect for the healthcare setting, instills confidence in patients, and showcases a commitment to the role. However, building a wardrobe suitable for the medical office can be a financial challenge, especially for those just starting in the field. By developing a strategic budget plan... | **0.96279** |
| Human-Written | Loved this tour! I grabbed a groupon and the price was great. It was the perfect way to explore New Orleans for someone who'd never been there before and didn't know a lot about the history of the city. Our tour guide had tons of interesting tidbits about the city, and I really enjoyed the experience. Highly recommended tour. I actually thought we were just going to tour through the cemetery, but she took us around the French Quarter for the first hour, and the cemetery for the second half of the tour. You'll meet up in front of a grocery store (seems strange at first, but it's not terribly hard to find, and it'll give you a chance to get some water), and you'll stop at a visitor center part way through the tour for a bathroom break if needed. This tour was one of my favorite parts of my trip! | **0.09172** |

Table 20: Representative examples of texts from various sources with their predicted probability of being AI-generated according to `DivEye`.

