# OpenReview forum: "Diversity Boosts AI-Generated Text Detection"
_TMLR — Accepted by TMLR_

### Review · Reviewer_onBt · 2025-12-03

**Summary Of Contributions:**

This paper proposes DivEye, a detection method that uses statistical patterns in token-level surprisal to distinguish human-written text from machine-generated text. Instead of relying on a fine-tuned neural classifier or a large LM as a detector, the approach extracts a compact set of surprisal-based features—covering distributional metrics (mean, variance, skewness, kurtosis) and temporal dynamics (first- and second-order differences, entropy, autocorrelation)—and feeds them into a lightweight classifier. The core idea is that human text tends to exhibit richer and more irregular surprisal fluctuations, whereas LLM-generated text often reflects smoother and more predictable patterns.

The empirical evaluation is quite extensive. The method is tested across multiple datasets (MAGE, RAID, HC3), various generator families, adversarial paraphrasing settings, and multilingual scenarios. DivEye performs strongly as a zero-shot detector and consistently improves the accuracy of several existing detectors when used as an auxiliary signal. The experiments also examine different surprisal base models and include feature-importance analyses that help explain why the method works.

Overall, the paper is clearly written, the core idea is intuitive, and the results support the main claims.

**Strengths:**
- Conceptually simple and well-motivated approach
- Strong performance across diverse benchmarks
- Works in zero-shot settings and with small LMs
- Offers interpretability through surprisal features
- Complements and improves existing detectors

**Weaknesses:**
- Heavy reliance on hand-crafted features may affect long-term robustness
- Some inconsistency in out-of-distribution performance
- Surprisal is primarily computed using GPT-2, which may not capture modern LM behavior
- Missing analysis on sampling/temperature effects and text-length sensitivity
- Real-world, organically sourced text evaluation is limited

**Additional Comments:**

Overall, I found this to be a thoughtful and well-executed paper. The proposed approach sits at a nice intersection of simplicity, interpretability, and practical utility. The empirical results are extensive and demonstrate that surprisal-based diversity features are surprisingly effective at capturing differences between human and LLM text. I also appreciate the inclusion of ablation studies and multiple base models, as these help reveal which aspects of the method matter most.

With some additional discussion around limitations and the suggested clarifications, I believe the paper would be even stronger. But as it stands, it already makes a solid and useful contribution to the area of machine-generated text detection.

**Audience:**

Yes

**Audience Explanation:**

AI-generated text detection is an active and increasingly important area, and the community is still exploring methods that are reliable in open, real-world settings. A technique that is zero-shot, computationally efficient, and interpretable is relevant to both researchers and practitioners. In particular, the idea of analyzing surprisal dynamics rather than solely average likelihood is interesting and may inspire follow-up work. The ability to “boost” existing detectors without retraining is also a practical contribution that many TMLR readers would appreciate.

**Broader Impact Concerns:**

I did not identify any major unaddressed ethical concerns. The intended purpose—detecting AI-generated text to mitigate misinformation, academic misconduct, and similar risks—is aligned with positive societal outcomes. However, as with all detection systems, there are a few considerations worth emphasizing:

- **False positives** can disproportionately affect non-native speakers or individuals with atypical writing styles.
- **Overreliance** on automated detectors by institutions may lead to unfair outcomes if results are treated as definitive.
- **Adversarial dynamics** are inherent to this field; stronger detectors may motivate more sophisticated evasion strategies.

These points do not detract from the contribution but should be acknowledged explicitly when deploying such systems in high-stakes environments.

**Claims And Evidence:**

Yes

**Claims Explanation:**

The evidence presented in the paper is generally strong and well-organized. The authors support their hypothesis with an initial empirical study showing meaningful differences in surprisal behavior between human and GPT-4-Turbo outputs, which helps ground the motivation. The experimental section covers a wide range of benchmarks—including both adversarial datasets and multilingual settings—and DivEye consistently performs at or near the top among zero-shot detectors. The improvement it provides when integrated with other detectors also reinforces the idea that it captures a complementary signal.

There are some areas that could benefit from additional clarification (e.g., how performance changes with different decoding temperatures or with very short inputs), but these do not undermine the validity of the main claims. Overall, the evidence is convincing and supports the conclusions drawn.

**Requested Changes:**

**Important (but not blocking)**
1. Clarify the role of the base LM used for surprisal.
    The method primarily uses GPT-2, which is quite outdated relative to the generators being evaluated. A deeper discussion of mismatch  scenarios (e.g., when the surprisal model is much weaker than the generator) would help practitioners understand the limits of the approach.

2. Provide more insight into out-of-distribution failure cases.
In some OOD testbeds, DivEye does not always outperform all baselines. Even a brief analysis of which text types or domains present difficulty would strengthen the generalization claims.

3. Discuss the effect of decoding strategies.
Since surprisal variance is central to the method, different sampling temperatures or top-p values could meaningfully affect detectability. A small study or at least a discussion would be helpful.

4. Consider a short analysis of text-length sensitivity.
Many real-world inputs are short (e.g., tweets, messages). Because the method aggregates statistics across tokens, performance likely varies with sequence length; understanding that behavior would be useful.



**Suggestions / Nice-to-have**

5. Add some evaluation on organically collected human text.
The existing benchmarks are strong but controlled. Even a small qualitative evaluation on real-world short-form text, essays, or forum posts would make the practical value clearer.

6. Minor clarity improvements in figures and captions.
A few figures are dense; slightly expanding captions or referencing them more explicitly in the text would improve readability.

---

> ### Comment · Reviewer_onBt · 2025-12-15
> **Questions on edge cases and surprisal model assumptions**
>
> Thanks to the authors for the clear and well-executed submission.
>
> I generally agree with the other reviewers assessments regarding the strong empirical performance, simplicity, and interpretability of the approach, particularly its usefulness as a lightweight signal that can complement existing detectors.
>
> One aspect I would like a bit more clarity on is how robust the core hypothesis is in certain edge cases. In particular, it would be helpful to better understand how DivEye behaves on text written by non-native speakers or on highly formal or templated human writing (e.g., legal or technical documents), where surprisal patterns may be more regular and potentially closer to model-generated text.
>
> I would also appreciate a brief discussion clarifying the role and limitations of using a relatively weak frozen LM (such as GPT-2) as the surprisal model, especially in cases where the generator being evaluated is substantially stronger. Some guidance on how to interpret results in these mismatch scenarios would help practitioners better understand the scope of the method.

---

> ### Author Response · Authors · 2025-12-18
> **Rebuttal by Authors (1/2)**
>
> We sincerely thank the reviewer `onBt` for their feedback. We appreciate the recognition of the novelty and potential impact of our work. The reviewer's comments have helped us further clarify and strengthen the contributions of DivEye, and we address each point in detail below.
>
> ---
>
> ### W1. Long-term robustness
> We appreciate the reviewer's feedback regarding the potential brittleness of hand-crafted features in the face of rapidly evolving LLMs. While it is true that LLMs are becoming increasingly fluent, our method targets a fundamental objective function of these models (likelihood maximization) which persists regardless of architectural improvements.
>
> To empirically validate the long-term robustness of our approach, we conducted a robustness test where we utilized an outdated evaluator model (GPT-2) to detect text generated by modern SoTA LLMs.
>
> Specifically, we generated a dataset of 500 persuasive essays using a spectrum of models ranging from older architectures to the most recent high-performance models.
>
> | Models | AI Accuracy |
> | --- | --- |
> | GPT-5o-latest (2025) | 98.6% |
> | Claude-4.5-Sonnet (2025) | 94.8% |
> | o1 (from OpenAI) (2024) | 97.4% |
> | Mistral-7B-v0.1 (2023) | 96.6% |
> | Llama-2-7B (2023) | 93.6% |
> | GPT-2-XL (2019) | 98.8% |
>
>
> Despite the significant capability gap between the observer (the frozen GPT-2 evaluator) and the generator, DivEye maintained high detection accuracy across all generations. We hope that this trend would continue for future models as well.
>
> ---
>
> ## Discussion about Changes
>
>
> ### W3 & C1. Base LM in DivEye
> We appreciate the opportunity to clarify the role of the base LM in our framework. The choice to primarily use GPT-2 was a deliberate design decision: by employing a significantly weaker and outdated observer model to detect text generated by SoTA generator models, we aimed to demonstrate that the temporal information about unpredictability is a fundamental statistical signature of AI text, rather than an artifact dependent on a matched model capability.
>
> Our results in Section 4.6 and Appendix D.3 confirm this hypothesis: despite the capacity mismatch, GPT-2 effectively captures the necessary signal to distinguish human from machine text. Furthermore, we extensively explored the impact of using stronger base models (Llama-3.1-8B, Falcon-7B) in Figure 6a and Table 6, showing that while a larger observer model does yield performance gains, the lightweight GPT-2 baseline already provides highly competitive results.
>
> Thus, while we recommend researchers utilize more capable models for production environments to maximize detection accuracy, our findings with GPT-2 establish a rigorous lower bound on DivEye's performance. This mismatch is a feature rather than a limitation, demonstrating that DivEye's efficacy persists even when the detector is significantly weaker than the generator. We state all of these findings in Appendix J.
>
> ---
>
> ### W2 & C2. Inconsistency in OOD performance
>
> We appreciate the reviewer's suggestion to critically examine this inconsistency.
>
> We conduct an experiment using MAGE's Testbed 7 (meant for OOD text detection): unseen domains generated by an unseen SoTA model (GPT-4). Our analysis reveals that DivEye's performance dips specifically in contexts like IMDb and DialogSum. These domains are characterized by shorter or highly informal texts (movie reviews, dialogue turns) which often lack the sustained length required for DivEye's features to fully stabilize.
>
> As noted in our sensitivity analysis below, the statistical signature of human texts is most distinct in longer continuous texts; consequently, the high brevity of these samples narrow the distributional gap between human and GPT-4 generation. However, despite these absolute drops in accuracy compared to in-distribution results, DivEye still outperforms most existing baselines on this testbed, proving that its core hypothesis remains a more reliable discriminator than raw perplexity even in these challenging cases.
>
> ---

---

> ### Author Response · Authors · 2025-12-18
> **Rebuttal by Authors (2/2)**
>
> ### W4 & C3. Decoding Strategies
> We agree that since DivEye relies on surprisal variance, and therefore analyzing the impact of decoding strategies is crucial to ensure the method's stability. Regarding sampling temperature, we refer the reviewer to Appendix E.3 of our paper, where we have already conducted an in-depth analysis. Specifically, we tested DivEye across a wide range of temperatures ($T \in [0.1-1.6]$) in both intra-model (GPT-2) and cross-model (Llama-3.1-8B) settings. As shown in Tables 16 and 17, DivEye exhibits remarkable robustness, maintaining high AUROC scores and detection accuracy even at extreme temperatures like $T=1.6$, where generation diversity is maximized.
>
> To complement the temperature analysis, we provide a specific evaluation of top-$p$ sampling here. We tested DivEye on 100 texts (per $p$) generated (by Llama-3.1-8B) with varying top-$p$ thresholds $p \in \{0.1, 0.3, 0.5, 0.8, 0.95\}$ to assess sensitivity.
>
> | Top-$p$ | AI Accuracy |
> | --- | --- |
> | $p=0.1$ | 96% |
> | $p=0.3$ | 94% |
> | $p=0.5$ | 97% |
> | $p=0.8$ | 93% |
> | $p=0.95$ | 95% |
> | **Average** | **95%** |
>
>
> Our experiments show that DivEye's performance remains consistent across these values, with negligible variation in detection accuracy. This stability arises because while top-$p$ sampling removes low-probability tokens, it does not introduce the "bursty" structural unpredictability characteristic of human writing; the remaining high-probability tokens still follow a AI-like surprisal trajectory that DivEye effectively detects. We have included these results in Appendix E.5.
>
> ---
>
> ### C4. Text Sensitivity
> We agree that understanding the impact of sequence length is critical for real-world deployment on short inputs. To quantify this behavior, we conducted an experiment where we systematically truncated input texts to varying lengths (ranging from 10 to 256 tokens) and evaluated DivEye's detection performance at each interval.
>
> Our results (Figure 14) demonstrate a clear scaling law between input length and detection accuracy. While performance is naturally constrained on extremely short fragments (0.62 AUROC at 10 tokens) due to the limited statistical support, DivEye rapidly stabilizes as context increases, achieving a robust 0.88 AUROC by 256 tokens. This confirms that while this statistical signature of human writing requires a minimum context window to emerge, it becomes reliably detectable within standard paragraph lengths.
>
> We have included these detailed results in Appendix D.9.4 and explicitly discuss the lower-bound limitations of our method on short texts in Appendix J.
>
> ---
>
> ### C5. Evaluation of Human Text
> We appreciate the reviewer's suggestion to demonstrate practical value through real-world examples. We would like to clarify that our evaluation already incorporates naturally sourced text; specifically, the experiments in Appendix D.5 utilize human-authored texts drawn directly from diverse, non-synthetic sources such as arXiv abstracts and Yelp reviews.
>
> To further reinforce this with a concrete qualitative snapshot, we conducted an additional evaluation on 50 randomly sampled arXiv abstracts and 50 random Reddit answers from the r/ELI5 subreddit. All texts have been taken from before 2011.
>
> DivEye achieved 100% accuracy on the arXiv samples and 96% accuracy on the ELI5 samples (which is not apart of any dataset), successfully identifying them as human-authored. These results on this data reinforce that DivEye's reliance on rhythmic surprisal patterns translates effectively to uncontrolled, organic writing environments.
>
> ---
>
> ### C6. Further Clarifications
>
> We appreciate the reviewer's ethical considerations and have explicitly discussed these broader concerns, including potential biases against non-native speakers and the risks of adversarial dynamics, in Appendix J. Moreover, we highly appreciate the reviewer's keen eye and will update the captions of figures to be more elaborate and detailed in the camera-ready version.
>
> Regarding the reviewer's interest in edge case robustness and for a detailed analysis of DivEye's robustness on non-native speakers and highly formal writing, please refer to our response (W2 & C1) to Reviewer `4ge7`, where we present specific experimental results demonstrating that the core hypothesis holds in these constrained domains.
>
> ---
> We appreciate the reviewer’s thoughtful engagement and hope that this discussion clarifies all concerns.

---

### Review · Reviewer_4ge7 · 2025-12-04

**Summary Of Contributions:**

This paper introduces DivEye, a framework for detecting AI-generated text based on the hypothesis that human-written text exhibits greater variability in token-level surprisal (the log probability of the current token based on pass texts) compared to LLM outputs. The method extracts nine statistical features, including surprisal's distributional moments, statistics for first and second-differences. The authors train a binary classifier (XGBoost based) using these features. They also experimented with combining this feature vector with existing detector's output as a booster.
Experiments with MAGE, RAID and HC3 show strong performance, robustness on adversarial attacks and multilingual texts, and good computational efficiency.

Strengths
- The underlying assumption - human writing has richer variability in unpredictability than LLM outputs, is well motivated, and nicely illustrated by the distribution plots in Figure 2.
- The DivEye features representing the surprisal distributions is clearly explained. Combined with the experiments performances and the feature importance analysis, show that it is simple, intuitive yet effective design. They are also easy and cost-efficient to generate - require a single forward pass through a small model + lightweight statistics computations.
- The authors perform thorough experiments on multiple benchmarks, domains, generator models, adversarial attack types.
- Overall, it is a simple yet effective framework with good interpretability and practicality. With highlights of model-agnostic, low computational cost, and potential of serving as booster to existing detectors.

Weakness
- The novelty is limited. The DivEye features are standard statistics applied to a known signal (surprisal) - moments of a distribution and basic time-series statistics. The paper does not justify why these features sufficiently describes of the surprisal distributions, and the actual gain comparing to the statistics used in existing frameworks, which already discussed entropy, curvature methods, etc.
- Potential brittleness of the core hypothesis. The assumption that human text is more creative and unpredictable may break in several scenarios, including non-native speakers who may use simpler vocabulary/grammar and more predictive structures, and formal/legal writing which is highly templated with strict choice of wordings.

**Audience:**

Yes

**Audience Explanation:**

AI-test detection will be an increasing important challenge, with growing difficulties as the LLMs getting better. The approach is simple, interpretable, resource-light, and can be combined with existing detectors. It could be easily tested and adopted by interested readers.

**Broader Impact Concerns:**

The most important consideration will be wrongful accusations - as mentioned above, non-native speakers may use less diverse writing style, and their writings have been falsely classified to AI-generated texts with higher rates. The paper should include impact and guidelines on using DivEye.

**Claims And Evidence:**

Yes

**Claims Explanation:**

The claims are largely supported. On MAGE, DivEye attains top zero-shot AUROC/AvgAcc across ID and OOD testbeds (Table 1). On RAID, it nearly ties the best fine-tuned open baseline in AUROC (0.984 vs 0.986) and clearly beats other zero-shot methods (Table 2). Its robustness is further shown by adversarial transformations. The fact it further boosted existing detector's result show that it captures some orthogonal signals.

**Requested Changes:**

- Bias & false-positive analysis. Evaluate DivEye's performances on edges cases, in particular, non-native English human writings, and highly-professional and formal writings such as legal documents. (Critical)
- Provide a more thorough theoretical/empirical decomposition that explains why the DivEye features add signal beyond the statistics captured by the existing models. Provide a head-to-head comparisons with identical evaluator LM - swap DivEye features out for the existing ones(entropy/log-rank/etc) computed from the same evaluator LM and run identical pipeline. (Critical).
- Since core idea of DivEye is surprisal feature engineering, it would be nice to include further feature importance and significance analysis, such as leave-one-out/leave-group-out tests with error bars, SHAP, permutation importances. (Strengthening)
- Provide more feature visualisations, such as distributions on various text types (including the edge cases). (Strengthening)

---

> ### Author Response · Authors · 2025-12-18
> **Rebuttal by Authors (1/2)**
>
> We sincerely thank the reviewer `4ge7` for the feedback and the positive evaluation of our work. We also acknowledge the limitations highlighted by the reviewer below:
>
> ---
>
> ### W1 & C2. Novelty of DivEye
> We would like to thank the reviewer for their critique regarding the novelty of our features. While we agree that the individual mathematical operators are standard statistical tools, our novelty lies not in the operators themselves, but in their application to a previously underutilized signal: the temporal evolution of surprisal.
>
> Existing zero-shot frameworks, such as Entropy, LogRank, and FastDetectGPT predominantly rely on static probability distributions or local curvature snapshots.
>
> In contrast, DivEye is theoretically grounded in the hypothesis that AI generation, optimized for likelihood maximization, produces statistically homogeneous sequences. The selection of these specific features is not arbitrary but is justified by the necessity of capturing this temporal structure, a claim we validate empirically in Appendix D.6 and Table 8. We also ground this claim with linguistic literature in Section 3.1.
>
> We find that second-order entropy is the single most impactful feature in our framework. Removing this temporal feature results in the largest statistically significant drop in AUC (-0.0263, p=0.001). This demonstrates that the standard distributional moments used in prior work are insufficient on their own to fully characterize the signal.
>
> Furthermore, regarding the actual gain compared to existing frameworks, our results demonstrate that DivEye captures a strictly orthogonal signal as seen in Table 4. To explicitly quantify the boosting power of DivEye compared to standard statistics, we performed an additional experiment (on MAGE, Testbed 4) where we evaluated the performance gain of adding DivEye features to baseline detectors (i.e. Entropy, LogRank, and DetectLLM) using the same evaluator LM (GPT-2).
>
> |  Framework   | AUROC | $\Delta$ : Relative Boost |
> | --- | --- | --- |
> | Entropy     | 0.67    | - |
> | LogRank     | 0.68    | - |
> | DetectLLM   | 0.75    | - |
> | DivEye      | 0.88    | - |
> | DivEye + Entropy     | 0.89    | 32.83% |
> | DivEye + LogRank     |  0.9   | 32.35% |
> | DivEye + DetectLLM    | 0.9    |12.96% |
>
>
> If DivEye were simply replicating the statistics of curvature or entropy methods, combining them would yield diminishing returns (proving that DivEye adds signal beyond existing detectors). Additionally, the feature importance analysis in Figure 10 reveals that DivEye retains dominant influence when ensembled with established detectors.
>
> ---
>
> ## Discussions about Changes
>
> ### W2 & C1. Robustness across Non-Native Writing and Professional Domains
> We acknowledge the reviewer's valid concern that reliance on unpredictability metrics could lead to higher false-positive rates in text types that are naturally more simple, such as non-native writing or highly formal documents. To rigorously test our core hypothesis, we conducted two targeted experiments on these specific edge cases.
>
>
> **Performance on Non-Native Writings**
> To evaluate potential bias against non-native speakers, we tested DivEye on the COREFL dataset [1]. This dataset (consisting of 1426 samples) includes essays written by German and Spanish native speakers and categorizes them by English proficiency levels.
>
> | Framework | Average Accuracy |
> | --- | --- |
> | Binoculars | 68.34% |
> | FastDetectGPT | 71.71% |
> | DivEye | 82.07% |
>
> Crucially, as illustrated in Figure 12, our granular analysis reveals that performance of all frameworks correlates with linguistic complexity: A1 texts (the most beginner class) are the most difficult to detect, while accuracy steadily improves as proficiency rises. As for DivEye, this suggests that detection becomes easier as the text exhibits the richer rhythmic signatures of advanced writing. We present these more granular results in Appendix D.9.1.
>
> ---
>
> **Performance on Professional Documents**
> To address the concern about highly professional writing, we evaluate DivEye on the LEDGER dataset [2], which is a corpus of legal provisions in contracts.
>
> | Framework | Accuracy |
> | --- | --- |
> | Binoculars | 91.67% |
> | FastDetectGPT | 94.8% |
> | DivEye | 99.49% |
>
> Despite the rigid structure of formal writing, DivEye outperforms other frameworks and successfully identifies human-authored professional text. We have added these results to Appendix D.9.2.
>
> In conclusion, these targeted experiments demonstrate that DivEye's core hypothesis holds true. Whether applied to simplified non-native syntax or professional documents, DivEye effectively captures the statistical signals that traditional methods miss.
>
> ---

---

> > ### Author Response · Authors · 2025-12-18
> > **Rebuttal by Authors (2/2)**
> >
> > ### C3. Feature Importance and Analysis
> > We appreciate the reviewer's suggestion to further validate our feature engineering through significance testing. We would like to direct the reviewer to Appendix D.6, where we have already included a detailed quantitative analysis of feature importance and statistical significance. Specifically, Figure 11 visualizes the relative feature importance, demonstrating that temporal features such as the entropy of second derivatives contribute most significantly to the decision boundary.
> >
> > Furthermore, Table 8 presents a leave-one-out ablation study where we quantified the impact of removing each feature on AUC performance. To ensure statistical rigor, this analysis includes paired bootstrap tests yielding p-values, which confirm that the contribution of every feature in the DivEye vector is statistically significant ($p < 0.05$), with the removal of second-order entropy causing the largest performance drop.
> >
> > ---
> >
> > ### C4. Visualizing Writings
> >
> > We acknowledge the reviewer's request for visual insights, particularly regarding edge cases. In alignment with our response to W2 & C1, we have expanded our analysis to include distribution plots for the specific domains discussed.
> >
> > These visualizations are now available in Appendix D.9.3, where we illustrate the predicted probability distributions of DivEye across these challenging domains. While we previously presented distribution visualizations for MAGE Testbed 4 in Figure 3, we now plot MAGE Testbed 8, COREFL (Non-Native), and LEDGAR (Professional/Legal) for a comprehensive comparison. A curious observation is seen: LEDGAR proves to be the easiest to detect, exhibiting the strongest human signal. This reflects that even highly structured human writing retains distinct rhythmic signatures of surprisal that differentiate it sharply from AI generation, further validating our core hypothesis.
> >
> > ---
> >
> > We thank the reviewer for highlighting the broad impacts of DivEye. We fully share this concern and have explicitly addressed the ethical implications and other limitations in Appendix J.
> >
> > [1] Lozano et al. Designing and compiling a learner corpus of written and spoken narratives: COREFL. DOI:10.3726/978-3-653-05182-7. (2020).
> >
> > [2] Tuggener et al. LEDGAR: A Large-Scale Multi-label Corpus for Text Classification of Legal Provisions in Contracts. Proceedings of the Twelfth Language Resources and Evaluation Conference. (2020).

---

### Review · Reviewer_4Fu1 · 2025-12-09

**Summary Of Contributions:**

The submission tackles the problem of model-agnostic zero-shot detection of  AI-generated text. This refers to the challenge of detecting whether some text was generated by *any* AI system, rather than detecting text generated by a particular AI system.

The main contribution is the demonstration of the usefulness of particular features which capture distributional and temporal patterns in the per-token surprisal of the text, as extracted from a frozen LLM. The authors refer to these features as diversity statistics and hence call their classifier DivEye (I don’t understand the ‘Eye’ part of the name).

DivEye’s feature set consists of the first four moments of the distribution of surprisal values for a given piece of text, the mean and variance of the first-order difference of those surprisal values (like the first derivative, capturing the change in surprisal values over the course of the sequence), and the variance, entropy, and autocorrelation of the second order difference (the difference of differences) of the surprisal values. From these features they train a binary classifier on examples of human-written and AI-generated text. The performance of this classifier on its own and as a booster for existing detectors is characterised with several datasets and ablations.

The authors motivate these features by claiming that they capture aspects of natural human language that will be missing in LLM-generated text, given their training objective. They claim that human language shows greater stylistic diversity, resulting in variations and fluctuations in the predictability of tokens within a piece of text.

The idea of using first and second order difference features for sequence-based machine learning applications is not especially novel, but it is great to see them work so well in this particular application.

**Strengths:**
* The idea is simple, makes sense, and is clearly described
* SOTA or competitive performance on three benchmarks
* Thorough evaluation on several datasets including text generated by a variety of AI systems, several baseline comparisons, assessing in- and out-of-distribution generalization, efficiency, as well as multi-lingual, adversarial and base model robustness.


**Weaknesses:**
* Not always clear how exactly this approach relates to past work, which has also used features of the log probabilities of a frozen LLM for this task.
* The use of first and second order difference features is motivated by speculation about the nature of human language. These claims are presented as fact without reference to any of the relevant (e.g., linguistics) literature.
* The title does not clearly communicate what this paper is about
* Sometimes imprecise/confusing language: using ‘diversity’ and ‘unpredictability’ interchangeably, using “diversity statistics”/”metrics” as an umbrella term for distributional and temporal difference features.

**Audience:**

Yes

**Audience Explanation:**

Robust detection of AI-generated text in the wild is very much needed. The approach described here is easy to incorporate into existing approaches. These findings may also point to alternative LLM training objectives to capture precisely these features that currently distinguish human and AI text.

**Claims And Evidence:**

Yes

**Claims Explanation:**

Yes and No. The most important claims are supported by the empirical results of the paper. However, the authors motivate their approach with several claims about human language for which they cite no prior evidence.

I especially appreciate the appendices which show the importance of each of the DivEye features.

**Requested Changes:**

**Critical requested changes:**
* Figure 3: To make the figures easier to understand, figure legends should say “Human-written” and “AI-generated” instead of “True Label 0” and “True Label 1” and the x-axis label should say something to indicate that it shows how likely the classifier judges the text to be human-written
* The set of 9 features seems somewhat arbitrary. Why not include the mean of the second order difference or the entropy of the first order difference (or any of the other missing statistics)? If other features were also explored and these 9 were found to be the most informative or some other feature selection was performed, then that should be included.
* The title should be revised to more clearly communicate the contribution of the paper, which is focused on the inclusion of distributional and temporal features of surprisal.
* There is little to no discussion at the end of the main paper. Some space should be reserved at the end to discuss how the approach builds on and differs from what's been done before, discuss limitations of the approach and any avenues for future work.


**Optional suggested changes:**
* If the authors wish to motivate their features by reference to properties of human language (stylistic diversity, burstiness, rhythmicity, etc.), it would be good to cite some literature to support these claims.
* I found sections 1-3 somewhat repetitive. To make room for a lengthier discussion, you might find opportunities to make the intro + methods sections more concise. I don't see the need for a separate box for Remark 1: Proof sketch. This is not something to be proved analytically anyway.
* Modern LLMs in use today have been trained on a variety of objectives, not merely to maximize the overall likelihood of the generated text, which the authors focus on when motivating their approach. How do these additional objectives affect the success of the DivEye? For example, if you generate text using the self-supervised pretrained, supervised instruction-tuned, and RLHF-trained versions of the same model, does DivEye work best on the pretrained model? I wonder if these post-training steps would affect the importance of the different features tested here. Some discussion of pre-training vs post-training objectives could be included.

---

> ### Author Response · Authors · 2025-12-18
> **Rebuttal by Authors (1/2)**
>
> We would like to express our gratitude to Reviewer `4Fu1` for their thoughtful and constructive comments. We address each specific point in full detail below.
>
> ---
>
> ### W1. Relation to Prior Works
> We appreciate this observation and agree that distinguishing DivEye from prior methods is crucial.
>
> Methods like Entropy, LogRank, and DetectLLM typically rely on aggregate scalar metrics or static ranking distributions. By analyzing tokens in isolation, these approaches fundamentally overlook how unpredictability evolves sequentially across the text. In contrast, DivEye characterizes the temporal structure and distributional shape of the surprisal signal.
>
> As detailed in Section 3.2, we extract a feature vector that targets concrete signals missed by prior work: specifically, we calculate the first-order difference to measure the magnitude of immediate shifts in predictability between consecutive tokens, and the second-order difference to track the rate at which these shifts fluctuate. This allows us to distinguish between the predictability often found in LLM generations and the contrary nature of human writing, where periods of routine language are punctuated by sudden spikes of surprise.
>
> This distinction is also empirically validated by our ablation studies. The orthogonality of our features is demonstrated in Figure 10, where we integrated DivEye into ensembles with existing probability-based detectors. If DivEye were merely replicating the signal of methods like DetectLLM or the OpenAI Detector, its contribution would be redundant. Instead, DivEye retains dominant feature importance in these ensembles ($81$% with DetectLLM and $90$% with OpenAI Detector). This confirms that DivEye captures a unique structural signal, specifically the temporal evolution of unpredictability, that static likelihood-based detectors fundamentally miss.
>
> ---
>
> ### W2. Linguistic Motivation for DivEye's features
> We thank the reviewer for pointing out the need to ground our motivation in established linguistics literature.
>
> Our focus on the first and second-order derivatives of surprisal is not speculative but is directly motivated by the Uniform Information Density (UID) hypothesis. As Levy et al. [1] state in their foundational work: "If speakers behave optimally, they should structure their utterances so as to avoid peaks and troughs in information density." This principle establishes that the flow of surprisal is a fundamental structural constraint in language production, validating our decision to model surprisal as a dynamic series.
>
> However, while the UID hypothesis suggests a theoretical optimum, empirical linguistic work demonstrates that human production is highly nonstationary [2] and exhibits distinct "burstiness" in word recurrence [3]. Humans optimize locally, but topic shifts and semantic constraints cause rhythmic fluctuations that result in a unique structural signature.
>
> In contrast, LLMs often approximate the theoretical UID optimum far more strictly than humans do. Decoding strategies which maximize likelihood lead to unnaturally flat probability curves [4], devoid of the variance found in natural speech. DivEye's first-order and second-order features are thus designed to explicitly quantify this distinction: they measure this non-stationarity, distinguishing the organic, bursty rhythm of human writing from LLMs.
>
> We have updated Section 3.1 to explicitly cite these works.
>
> ---
>
> ### W3. Clarity about the Title
> We selected this title to capture the two central contributions of our work, though we are happy to refine it for greater precision.
>
> DivEye's features are fundamentally orthogonal to those used by existing state-of-the-art detectors. Because DivEye captures structural and temporal signals unseen by other models, it does not merely compete with them but actively complements them. As shown in our experiments, using DivEye as an auxiliary signal significantly boosts the performance of existing detectors (improving RADAR and others by up to 18.7%).
>
> To clarify the name: 'Div' refers to the diversity-based features central to our method. 'Eye' is simply a metaphor for this framework acting as an observer or detector that 'looks' at these features to spot AI text.
>
> ---

---

> > ### Author Response · Authors · 2025-12-18
> > **Rebuttal by Authors (2/2)**
> >
> > ## Discussions about Changes
> > ---
> >
> > ### C1. Why not more than 9 features?
> > We appreciate the reviewer's inquiry regarding the size of our feature set. Our decision to restrict DivEye to nine specific features was driven by a balance of computational efficiency and diminishing returns on performance, rather than an arbitrary cutoff. As suggested in Appendix J, the framework is flexible enough to accommodate additional features, but our empirical analysis suggests that adding more high-order moments yields negligible gains.
> >
> > We also perform a scaling experiment on these existing 9 features to prove that the AUROC tends to stagnate on adding the proposed mean and variance features. We also include other third and fourth-order features and these higher order features also suffered from similar issues.
> >
> > We have included all these findings in Appendix F.3 and Figure 15.
> >
> > ---
> >
> > ### C2. Pre-trained and Post-trained models
> > To address the reviewer's insightful question regarding the impact of post-training objectives (SFT, DPO, RLHF) on detectability, we conducted a controlled experiment using the Olmo-3-7B [5] model family. Since Olmo releases checkpoints for every stage of the training pipeline, it allows us to isolate the effect of the objective function while keeping the model architecture and size constant. We direct the reviewers to Figure 2 of the Olmo 3 paper [5] to understand the models used in this experiment.
> >
> > We evaluated DivEye's detection performance across eight distinct variants of the model (texts generated are from various domains, temperatures and top-p values), ranging from the base pre-trained checkpoint to advanced Thinking and RLVR-tuned versions. The results are summarized below:
> >
> > | Model Type | Training Phase | AI Accuracy |
> > | ---- | ---- | ---- |
> > |OLMo-3-7B-Base | Pre-trained Base | 97.33% |
> > |OLMo-3-7B-Instruct-SFT	| Instruct Version (finetuned by SFT) | 98.0% |
> > |OLMo-3-7B-Instruct-DPO	| Instruct Version (finetuned by SFT+DPO)| 100% |
> > |OLMo-3-7B-Instruct	| Instruct Version (finetuned by SFT+DPO+RLVR) | 98.67% |
> > |OLMo-3-7B-Thinking-SFT	| Thinking Version (finetuned by SFT) | 99.33% |
> > |OLMo-3-7B-Thinking-DPO	| Thinking Version (finetuned by SFT+DPO)| 100% |
> > |OLMo-3-7B-Thinking	| Thinking Version (finetuned by SFT+DPO+RLVR) | 100% |
> > |OLMo-3-7B-RL-Zero | Finetuned only by RLVR | 98.67% |
> >
> > Contrary to the intuition that aligned models might better mimic human stylistic nuances, we observe that alignment consistently enhances detectability. The pre-trained Base model yields the lowest (yet still robust) accuracy of 97.33%, whereas models fine-tuned with DPO and Thinking capabilities achieve perfect detection rates.
> >
> > This trend suggests that post-training interventions constrain the model's output distribution to a highly structured subspace. This constraint likely dampens the natural entropic variance present in the pre-training data, making it easier for DivEye to distinguish post-trained models from genuine human writing. We discuss these results in further detail in Appendix F.4 and Figure 16.
> >
> > ---
> >
> > ### C3. Discussions and Future Work
> > We appreciate the reviewer's suggestion to include a discussion at the end of the main paper. We would like to note that we have already provided an extensive discussion regarding DivEye's limitations, broader impacts, and avenues for future work in Appendix J.
> >
> > ---
> >
> > ### C4. Other minor changes
> > We agree with the reviewer's suggestions to improve clarity. We have updated Figure 3 by replacing the numeric labels with 'Human-written' and 'AI-generated' and renaming the x-axis to 'Predicted Probability of Human Authorship' to better indicate the classifier's judgment. Additionally, we will streamline Sections 1–3 to reduce redundancy and improve clarity on wordings (specifically around diversity/unpredictability) in the camera-ready version. We have also revised Remark 1 and embedded it directly into the main text.
> >
> > ---
> >
> > [1] Levy et al. Speakers optimize information density through syntactic reduction. NeurIPS. 2006.
> >
> > [2] Piantadosi et al. Word lengths are optimized for efficient communication. PNAS. 2011.
> >
> > [3] Altmann et al. Beyond word frequency: Bursts, lulls, and scaling in the temporal distributions of words. arXiv:0901.2349. 2009.
> >
> > [4] Holtzman et al. The Curious Case of Neural Text Degeneration. arXiv:1904.09751. 2020.
> >
> > [5] AllenAI et al. Olmo 3. arXiv:2512.13961. 2025.

---

### Decision · Action_Editor_1wue · 2026-01-20

**Recommendation:** Accept as is

**Additional Comments:**

~~One reviewer (Reviewer 4ge7) suggested Reproducibility Certification to acknowledge the provided code link, the thorough evaluation with detailed explanations, and the newly added experiments according to the reviews. I agree with the judgement and would like to thank the reviewer for the reorganization.~~
(Reproducibility Certification does not apply to this paper since the award is for papers whose primary purpose is reproduction of other published work.)

**Audience:**

Yes

**Audience Explanation:**

All reviewers agree that robust detection of AI-generated text is an important task; and this paper presented a simple, well-motivated and useful approach that would benefit both researchers and practitioners.

**Claims And Evidence:**

Yes

**Claims Explanation:**

This is a well-written paper with thorough evaluation. I especially appreciate the productive discussion phase, in which the authors have made revisions to the paper and addressed almost all of the concerns / curiosities (including optional ones) from the reviewers, by newly added experimental results. This is a firm Yes.